# Polyunsaturated fatty acids inhibit a pentameric ligand-gated ion channel through one of two binding sites

Noah M Dietzen[1†], Mark J Arcario[1†], Lawrence J Chen[1], John T Petroff II[1], K Trent Moreland[1], Kathiresan Krishnan[2], Grace Brannigan[3,4], Douglas F Covey[1,2,5,6], Wayland WL Cheng[1]*

[1]Department of Anesthesiology, Washington University in St. Louis, St Louis, United States; [2]Department of Developmental Biology, Washington University in St. Louis, St Louis, United States; [3]Center for the Computational and Integrative Biology, Rutgers University, Camden, United States; [4]Department of Physics, Rutgers University, Camden, United States; [5]Department of Psychiatry, Washington University in St. Louis, St. Louis, United States; [6]Taylor Institute for Innovative Psychiatric Research, Washington University in St. Louis, St. Louis, United States

**Abstract** Polyunsaturated fatty acids (PUFAs) inhibit pentameric ligand-gated ion channels (pLGICs) but the mechanism of inhibition is not well understood. The PUFA, docosahexaenoic acid (DHA), inhibits agonist responses of the pLGIC, ELIC, more effectively than palmitic acid, similar to the effects observed in the $GABA_A$ receptor and nicotinic acetylcholine receptor. Using photo-affinity labeling and coarse-grained molecular dynamics simulations, we identified two fatty acid binding sites in the outer transmembrane domain (TMD) of ELIC. Fatty acid binding to the photolabeled sites is selective for DHA over palmitic acid, and specific for an agonist-bound state. Hexadecyl-methanethiosulfonate modification of one of the two fatty acid binding sites in the outer TMD recapitulates the inhibitory effect of PUFAs in ELIC. The results demonstrate that DHA selectively binds to multiple sites in the outer TMD of ELIC, but that state-dependent binding to a single intrasubunit site mediates DHA inhibition of ELIC.

## Editor's evaluation

The authors use a combination of photolabeling and mass spectrometry to probe polyunsaturated fatty acid (PUFA) binding site locations in the pentameric ligand-gated ion channel (pLGIC) ELIC. The data strongly support the idea that DHA, but not PA, bind in the transmembrane domains of ELIC in two locations that overlap with that previously shown for the homolog GLIC. They also show that coarse-grained simulations can recapitulate the observation that DHA and not PA bind in this region, supporting the idea that such simulations can be useful for studying PUFA interactions with pLGICs. Strikingly, the authors provide evidence that DHA binding depends on the occupancy of the agonist site, which is an important observation that informs on molecular motions in the transmembrane domains in response to agonist binding. This work contributes to understanding the molecular underpinnings of PUFA modulation in pLGICs.

## Introduction

Fatty acids are major components of the cell membrane, and modulators of many ion channels including pentameric ligand-gated ion channels (pLGICs) (*Antollini and Barrantes, 2016*; *Fernández*

*For correspondence:
wayland.cheng@wustl.edu

†These authors contributed equally to this work

Competing interest: The authors declare that no competing interests exist.

*Nievas et al., 2008*; *Basak et al., 2017*; *Nabekura et al., 1998*; *Hamano et al., 1996*; *Minota and Watanabe, 1997*). Fatty acids inhibit neuronal pLGICs including the GABA$_A$ receptor (GABA$_A$R) (*Nabekura et al., 1998*) and nicotinic acetylcholine receptor (nAchR) (*Fernández Nievas et al., 2008*; *Hamano et al., 1996*; *Minota and Watanabe, 1997*; *Bouzat and Barrantes, 1993*), by reducing peak channel responses to agonist. Polyunsaturated fatty acids (PUFAs) have a stronger inhibitory effect than mono- or saturated fatty acids (*Hamano et al., 1996*; *Minota and Watanabe, 1997*). While the physiologic significance of PUFA inhibition of pLGICs is not fully understood, neuronal PUFAs affect neurologic processes in which pLGIC function is critical such as neurodevelopment (*McNamara and Carlson, 2006*) and cognition (*Hashimoto et al., 2017*). Neuronal PUFA content can change dramatically in pathologic conditions such as stroke and seizure (*Taha et al., 2009*), and PUFA inhibition of pLGICs under these conditions is likely an important determinant of neuronal excitability (*Antollini and Barrantes, 2016*; *Taha et al., 2013*).

Fatty acids modulate ion channel function allosterically by one of two general mechanisms: by direct binding to specific sites in the protein, or by altering the physical properties of the lipid bilayer and thereby indirectly impacting protein structure. There is evidence to support both mechanisms in different ion channels including pLGICs (*Cordero-Morales and Vásquez, 2018*; *Sogaard et al., 2006*). A direct mechanism is thought to be important in pLGICs based on a crystal structure of *Gloeobacter* ligand-gated ion channel (GLIC) in complex with docosahexaenoic acid (DHA), which showed a single binding site for DHA in the outer portion of the transmembrane domain (TMD) of this channel (*Basak et al., 2017*). However, whether fatty acids bind to other sites in pLGICs, and whether these sites are specific for PUFAs and mediate PUFA inhibition of channel activity are not established. It is possible that fatty acids bind to other sites in GLIC that were not resolved in the crystal structure due to greater flexibility of the fatty acid within those sites. Mutagenesis studies and molecular dynamics simulations also support a direct mechanism of fatty acid modulation of large conductance calcium-activated potassium channels (BK) (*Tian et al., 2016*) and voltage-gated potassium channels (Kv) (*Börjesson et al., 2008*; *Yazdi et al., 2016*), but biochemical identification of fatty acid binding sites in these and other ion channels remains a challenge.

Photo-affinity labeling is an alternative approach to identify lipid and small molecule binding sites in membrane proteins. Fatty acid photolabeling reagents have been used previously to identify fatty acid binding proteins (*Haberkant and Holthuis, 2014*; *Haberkant et al., 2013*; *Capone et al., 1983*), but not to map binding sites at the residue level. We introduce a new photolabeling reagent with optimal photochemistry coupled with intact protein and middle-down mass spectrometry (MS) (*Budelier et al., 2019*; *Budelier et al., 2017a*; *Cheng et al., 2018*) to identify fatty acid binding sites in the pLGIC, *Erwinia* ligand-gated ion channel (ELIC) (*Hilf and Dutzler, 2008*). Combining this approach with coarse-grained molecular dynamics (CGMD) simulations, we determine two fatty acid binding sites in the outer TMD of ELIC that are specific for DHA over palmitic acid when agonist is bound to the channel. Occupancy of one of these sites mediates the inhibitory effect of DHA on ELIC channel responses. The results argue for a direct mechanism of DHA inhibition of ELIC through state-dependent binding to a single site.

## Results
### DHA inhibits ELIC channel function
ELIC has been a useful model channel for investigating the structural determinants of lipid (*Hénault et al., 2019*; *Tong et al., 2019*) and anesthetic (*Kinde et al., 2016*; *Chen et al., 2015*; *Spurny et al., 2013*) modulation of pLGICs. We sought to determine if ELIC is a suitable model for examining the mechanism of fatty acid inhibition of pLGICs. ELIC was reconstituted in giant liposomes composed of a 2:1:1 ratio of POPC:POPE:POPG for excised patch voltage-clamp recordings and channel responses were evaluated by rapid application of the agonist, cysteamine, at 30 mM for 1 min. ELIC responses showed fast activation followed by a slower current decay or desensitization, consistent with previous reports of ELIC function in liposomes or HEK cells (*Tong et al., 2019*; *Laha et al., 2013*). The effect of fatty acids such as DHA (22:6) was determined by pre-applying fatty acid to the patch for 3 min followed by rapid application of cysteamine with the same concentration of fatty acid used in the pre-application (*Figure 1A*). Longer pre-application times with fatty acid did not yield a greater effect, indicating that 3 min was sufficient time to equilibrate the patch membrane with DHA from solution.

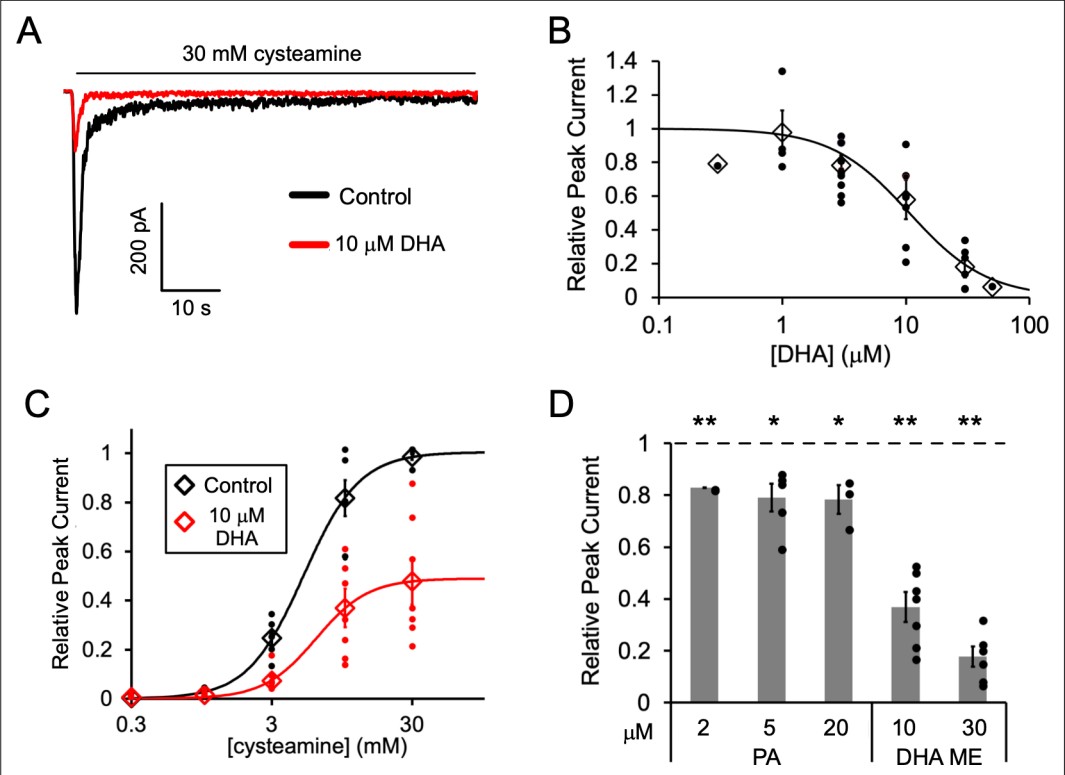

**Figure 1.** Fatty acid inhibition of ELIC. (**A**) Sample currents from excised patch-clamp (–60 mV) of WT ELIC in 2:1:1 POPC:POPE:POPG giant liposomes. Current responses are to 30 mM cysteamine before and after 3 min pre-application 10 µM DHA. (**B**) Peak current responses to 30 mM cysteamine normalized to control (absence of DHA) as a function of DHA concentration (n = 4–8, 0.3, and 50 µM n = 1,± SEM). The data are fit to a sigmoidal function yielding an $IC_{50}$ of 10.6 µM. (**C**) Peak current responses normalized to maximum response as a function of cysteamine concentration in the absence or presence of 10 µM DHA (n = 7,± SEM, control $EC_{50}$ = 5.5 ± 0.7 mM, 10 µM DHA $EC_{50}$ = 6.2 ± 0.6 mM). (**D**) Peak current responses to 30 mM cysteamine with 3 min pre-application of PA (palmitic acid) and DHA ME (DHA methyl ester) normalized to control (no lipid) (n = 4–6,± SEM). Statistical analysis of each sample was performed using a one-sample T-test against the control value of one (* p < 0.05, ** p < 0.01).

The online version of this article includes the following figure supplement(s) for figure 1:

**Figure supplement 1.** Inhibition of ELIC by DHA, PA and DHA ME.

Pre-application of DHA caused ELIC peak responses to decrease with an $IC_{50}$ of 10.6 µM (**Figure 1A and B**); there was almost complete inhibition of ELIC responses at 50 µM DHA. DHA also caused a ~ 50% reduction in steady state current relative to the peak current in the absence of DHA ($IC_{50}$ = 9.8 µM) and a similar percent reduction in the rate of current decay ($IC_{50}$ = 10.5 µM) (**Figure 1—figure supplement 1**). However, the $EC_{50}$ of cysteamine activation showed no significant change with the pre-application of 10 µM DHA (**Figure 1C**). Furthermore, pre-application with the saturated fatty acid, palmitic acid (PA, 16:0), significantly reduced ELIC peak responses to 30 mM cysteamine, but to a much lesser degree than DHA (**Figure 1D** and **Figure 1—figure supplement 1C**). 2, 5 and 20 µM PA yielded the same effect indicating that saturating concentrations of PA were used. Interestingly, the methyl ester of DHA (DHA ME) strongly inhibited ELIC similar to DHA (**Figure 1D** and **Figure 1—figure supplement 1D**), indicating that the carboxylate DHA headgroup is not required for this inhibitory effect. The effects of fatty acids on ELIC are qualitatively similar to the effects reported in the $GABA_AR$ (**Nabekura et al., 1998**; **Hamano et al., 1996**; **Sogaard et al., 2006**) and nAchR (**Minota and Watanabe, 1997**; **Vijayaraghavan et al., 1995**). PUFAs inhibit gating efficacy without shifting agonist potency, and DHA ME is just as effective as DHA (**Taha et al., 2013**). The results suggest that the mechanism of fatty acid inhibition of the $GABA_AR$ and nAchR is also present in ELIC.

One possibility for why DHA is a stronger inhibitor of ELIC than PA is that PA has a lower affinity for DHA binding sites in ELIC. Alternatively, the two fatty acids may bind to the same sites with similar affinity but different efficacy. To begin examining the interaction of fatty acids with ELIC, we performed CGMD on a single ELIC pentamer in a 2:1:1 POPC:POPE:POPG membrane containing either 4 mol% PA or DHA. Here, we used the MARTINI coarse-grained model, originally developed as a coarse-grained model for lipids, which represents four non-hydrogen atoms as a single coarse-grained bead with well-described properties (*Marrink and Tieleman, 2013*). This coarse-graining reduces the number of particles and associated calculations of intermolecular potential, and eliminates fast interatomic vibrational modes allowing larger timesteps. Together this leads to simulations at longer timescales. The cost of these modifications, however, is a lack of atomic resolution and limited sampling of protein conformational changes due to structural restraints required for protein models (*Alessandri et al., 2019*). Nevertheless, CGMD has been useful for sampling protein-lipid interactions by permitting more lipid diffusion over the simulation time scale (*Dämgen and Biggin, 2021*; *Sharp et al., 2019*). We first quantified lipid sorting using the boundary lipid metric, *B* (*Equation 2*), where B > 1 indicates enrichment compared to bulk membrane and $B < 1$ indicates relative depletion (*Sharp et al., 2019*). DHA shows significant enrichment in the lipids surrounding ELIC (B = 2.43 ± 0.16,± SEM, n = 4), while PA shows a relative depletion (B = 0.51 ± 0.02,± SEM, n = 4). To identify localized density that would indicate specific binding, we calculated the two-dimensional radial enhancement for each fatty acid species with ELIC centered at the origin (*Figure 2*). In these plots, white tiles represent no enrichment over expected bulk membrane density (radial enhancement, $\widetilde{\rho}_B = 1.0$); increasing enrichment is represented by deepening red tiles ($\widetilde{\rho}_B > 1.0$) and increasing depletion is represented by deepening blue tiles ($\widetilde{\rho}_B < 1.0$). It is immediately apparent that DHA shows several localized areas of high enrichment indicating specific sites of binding to ELIC, whereas PA shows areas of minimal enrichment that are more diffuse (*Figure 2*). Binding of DHA is observed mainly in two intrasubunit grooves located between M3 and M4 or M1 and M4. Moreover, these areas of localized binding are noted mostly in the outer leaflet (*Figure 2*, top right), suggesting DHA acts through a specific binding site in the outer leaflet of the membrane. These results are consistently demonstrated over multiple simulation replicates (*Figure 2— figure supplement 1* and *Figure 2—figure supplement 2*), indicating that in 2:1:1 POPC:POPE:POPG lipid membranes in which ELIC responses were measured, DHA binds to specific intrasubunit sites in the outer leaflet, which PA is not seen to strongly interact with. This supports the notion that DHA inhibits ELIC by binding to specific sites, while PA has low affinity for ELIC.

## Characterization of a fatty acid photolabeling reagent, KK242, in ELIC

To further identify fatty acid binding sites in ELIC, we used photo-affinity labeling coupled with MS. We began by testing a commercially available fatty acid photolabeling reagent, 9-(3-pent-4-ynyl-3 -H-diazirin-3-yl)-nonanoic acid or pacFA (Avanti Polar Lipids, *Figure 3A*). This bifunctional reagent contains a photo-reactive aliphatic diazirine in the alkyl tail and an alkyne for click chemistry (*Haberkant et al., 2013*). We photolabeled purified ELIC with 100 µM pacFA and analyzed the photolabeled protein by intact protein MS (*Budelier et al., 2017a*; *Cheng et al., 2018*; *Sugasawa et al., 2019*; *Budelier et al., 2017b*). Even at this high concentration, no photolabeling of ELIC by pacFA could be detected (*Figure 3B*).

We have previously shown that photolabeling reagents that contain an aliphatic diazirine favor labeling of nucleophilic amino acid side chains such as glutamate or aspartate (*Cheng et al., 2018*). Since the alkyl tail of pacFA, where the photo-reactive diazirine is located, is expected to interact with the hydrophobic regions of the ELIC TMD, one might expect that pacFA will not photolabel ELIC as there are no strong nucleophiles in the TMD facing the hydrophobic core of the membrane. In contrast to an aliphatic diazirine, a trifluoromethylphenyl-diazirine (TPD) shows less selectivity between amino acid side chains and offers favorable photochemistry for photolabeling studies with lipids (*Cheng et al., 2018*; *Budelier et al., 2017b*). Thus, we synthesized a fatty acid analogue photolabeling reagent, KK-242, which contains an ether-linked TPD group in the fatty acid alkyl tail and approximates the molecular dimensions of PA (~20 Å in length) (*Figure 3A*). Intact protein MS of ELIC photolabeled with 100 µM KK-242 showed two additional peaks corresponding to the mass of an ELIC subunit with one and two KK-242 labels (labeling efficiency ~15%) (*Figure 3B*). This result indicates that KK-242 labels ELIC efficiently, and that there is a minimum of two non-overlapping labeled sites per subunit.

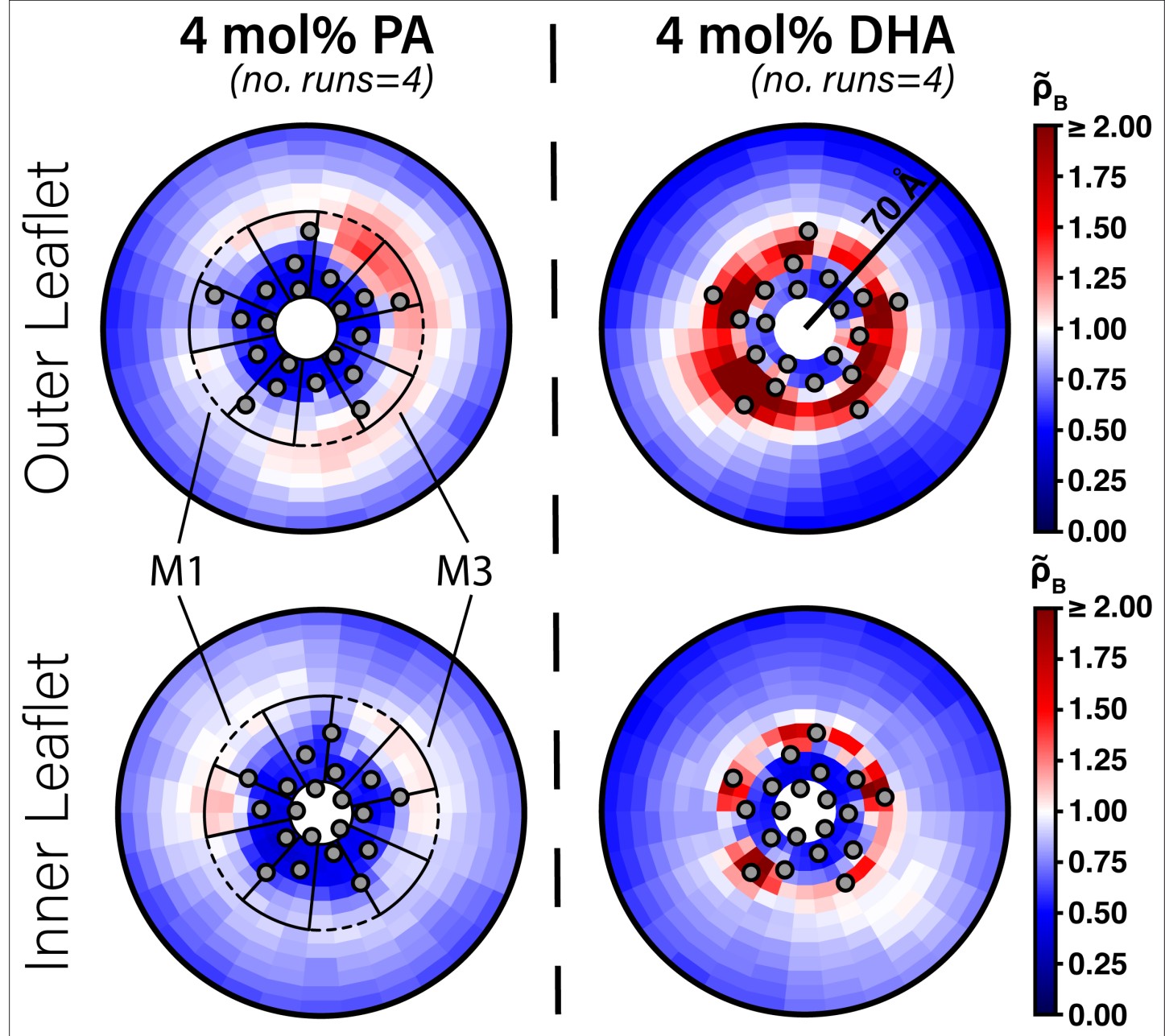

**Figure 2.** Enrichment of PA and DHA in ELIC. Two-dimensional enrichment plots for the 4 mol% PA (left column) and 4 mol% DHA (right column) simulation conditions are shown for fatty acid species within 70 Å of the ELIC pore. Separate enrichment for the outer (top row) and inner (bottom row) leaflet are plotted with the color bar at right demonstrating relative enrichment values compared to bulk membrane. $\tilde{\rho}_B > 1.0$ indicates enrichment over bulk membrane and $\tilde{\rho}_B < 1.0$ indicates relative depletion compared to bulk membrane (**Equation 4**). Data presented in this figure represent the average across four simulation runs. The gray circles in each plot represent the location of the transmembrane helices relative to the ELIC pore. The dashed black outline sectors demonstrate the boundaries of the M1 site as used for density affinity threshold calculations, whereas the solid black outline sectors demonstrate the boundaries of the M3 site.

The online version of this article includes the following figure supplement(s) for figure 2:

**Figure supplement 1.** Interaction of PA with ELIC.

**Figure supplement 2.** Interaction of DHA with ELIC.

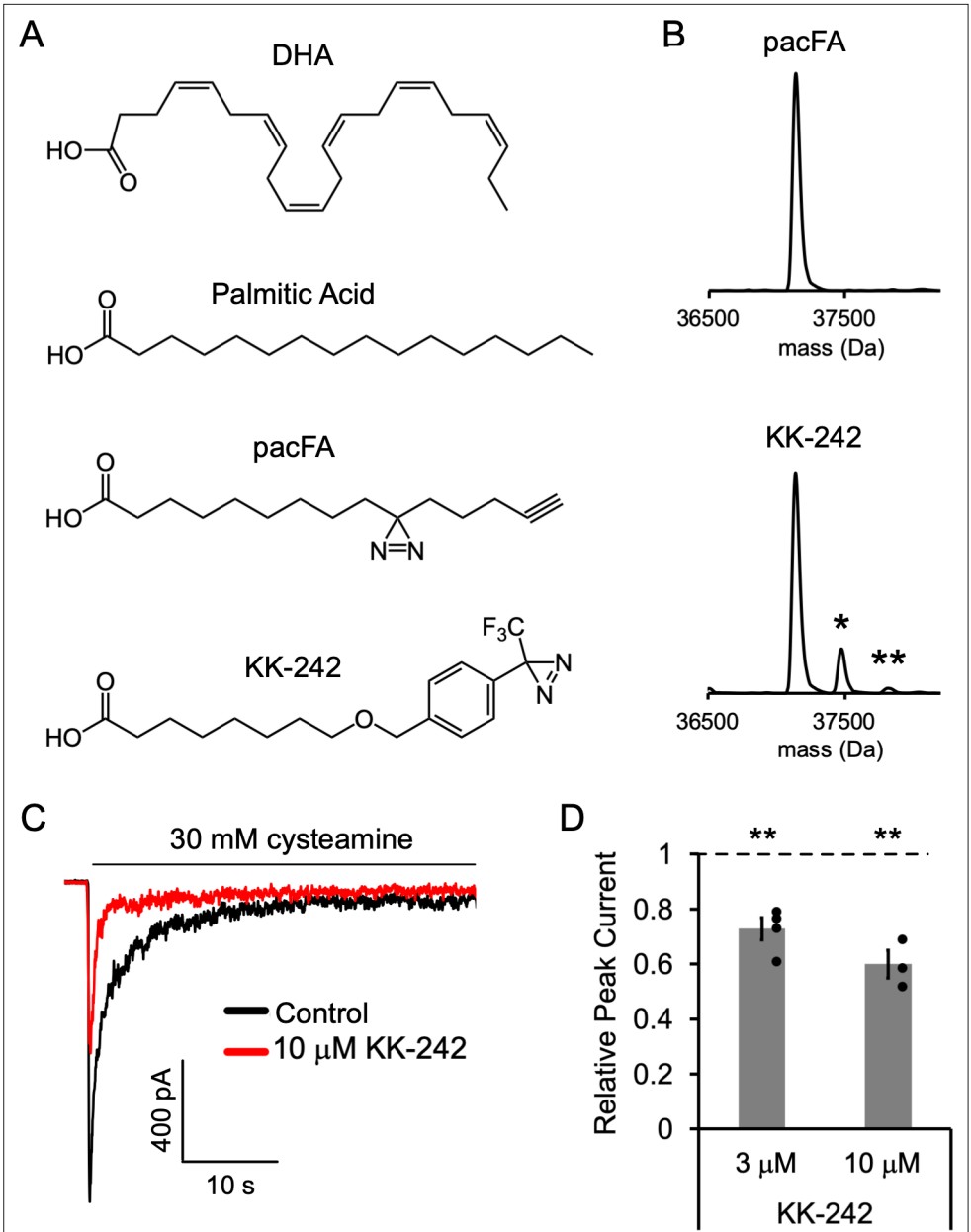

**Figure 3.** Fatty acid reagent photolabeling and inhibition of ELIC. (**A**) Chemical structures of DHA, PA, pacFA, and KK-242. (**B**) Deconvoluted intact protein mass spectra of WT ELIC photolabeled with 100 µM pacFA and 100 µM KK-242. In both spectra, the highest intensity peak corresponds to the WT ELIC subunit, and in the KK-242 spectrum, the peaks labeled * and ** indicate an ELIC subunit with 1 and 2 KK-242 adducts, respectively. (**C**) Sample currents from excised patch-clamp (–60 mV) of WT ELIC in 2:1:1 POPC:POPE:POPG giant liposomes. Current responses are to 30 mM cysteamine before and after 3 min pre-application of 10 µM KK-242. (**D**) Peak current responses to 30 mM cysteamine normalized to control (absence of KK-242) as a function of KK-242 concentration (n = 4,± SEM). Statistical analysis of each sample was performed using a one-sample T-test against the control value of one (** $p < 0.01$).

The online version of this article includes the following figure supplement(s) for figure 3:

**Figure supplement 1.** KK-242 inhibition of ELIC.

We next verified that this fatty acid analogue also inhibits ELIC function. 3 and 10 µM KK-242 inhibited ELIC peak responses with 10 µM showing a slightly greater effect (*Figure 3C and D*). The maximal efficacy of KK-242 could not be determined confidently because higher concentrations were insoluble in the aqueous solutions used for patch-clamp measurements. However, these results suggest that the efficacy of KK-242 is slightly higher than PA and lower than DHA. KK-242 also modestly decreased relative steady state current and the time constant of current decay, although these differences were not statistically significant (*Figure 3—figure supplement 1*). Thus, KK-242 is a suitable fatty acid analogue photolabeling reagent, qualitatively reproducing the inhibitory effects of fatty acids on ELIC responses.

## DHA binds to two sites in ELIC in an agonist-bound state

To identify the photolabeled sites of KK-242 in ELIC, we analyzed ELIC photolabeled with 100 µM KK-242 with and without 30 mM cysteamine using a tryptic middle-down MS approach (*Cheng et al., 2018*). Initial analysis of the LC-MS/MS data using PEAKS showed 100% coverage of the ELIC sequence, including high intensity peptides for all four transmembrane helices and KK-242 modified peptides encompassing M3 and M4 (*Figure 4—figure supplement 1*). Both photolabeled peptides had a longer retention time than the corresponding unlabeled peptide, as would be expected with reverse phase chromatography (*Figure 4—figure supplement 2*), and high mass accuracy (*Figure 4—source data 1*). No photolabeled peptides were identified in the extracellular domain, M1 or M2.

The labeling efficiencies of ELIC subunits at 100 µM KK-242 with and without 30 mM cysteamine were indistinguishable by intact protein MS (~15%). Labeling efficiencies of M4 and M3 were also measured from the middle-down MS data. This was defined as the ratio of MS intensities of the labeled peptide to the sum of labeled and unlabeled peptides. MS intensities depend on peptide abundance and ionization efficiency. Since the ionization efficiency may differ between unlabeled and labeled peptides, we call these measured values apparent labeling efficiencies. KK-242 apparent labeling efficiencies of M4 and M3 with and without cysteamine, and the corresponding fragmentation spectra were not significantly different (*Figure 4—figure supplement 3*). Thus, we focused on the fragmentation spectra of photolabeled ELIC in the presence of cysteamine. Manual analysis of the MS2 spectra of both photolabeled peptides revealed fragment ions containing the KK-242 mass (*Figure 4A and B*). For the M3 peptide, three *b* ions localize the photolabeled residue to Q264 (*Figure 4A*). For the M4 peptide, a series of *b* and *y* ions indicate two photolabeled residues: C313 and R318 (*Figure 4B*). In both peptides, the same fragment ions with and without the KK-242 mass are frequently present most likely representing partial neutral loss of KK-242. While the possibility of 100% neutral loss raises some uncertainty about the residues that are photolabeled, the spectra are most consistent with photolabeling of Q264 in M3, and C313 and R318 in M4. Q264 is in M3 on one side of M4, while C313 is facing the other side of M4 adjacent to M1 (*Figure 4A and B*). Most structures of ELIC do not resolve R318 at the C-terminal end of M4; however, a recent cryo-EM structure of ELIC shows this residue facing M3 adjacent to Q264 (*Kumar et al., 2021*). To test the plausibility that KK-242 can photolabel these residues, we docked KK-242 in two volumes that encompass the photolabeled residues and the outer interfacial region of the ELIC TMD. The docked poses were examined after clustering at 5 Å RMSD. While many distinct binding poses were obtained (KK-242 has 13 rotatable bonds), only a single cluster from each site show poses where the diazirine is closest to the photolabeled residue (*Figure 4A and B*). When this is the case, the carboxylate group of KK-242 is approximately at the interfacial region of the membrane (*Figure 4A and B*). Other clusters were obtained at each site showing poses that are less plausible for a fatty acid analogue in a lipid bilayer or detergent micelle (*Figure 4—figure supplement 4A* and B). Thus, KK-242 photolabels two distinct sites, and the photolabeled residues and docking results suggest that these sites are on either side of the outer portion of M4 (*Figure 4—figure supplement 4C*). These sites are analogous to the sites identified in the GLIC crystal structure where a phospholipid was bound to the side of M4 facing M1 and DHA was bound to the side of M4 facing M3 (*Basak et al., 2017*).

To test whether KK-242 photolabeled sites are specific binding sites for DHA or PA, a competition experiment was performed: apparent labeling efficiencies at 10 µM KK-242 were evaluated in the presence of 30 mM cysteamine with and without the addition of DHA or PA. At this lower concentration of KK-242, labeling of M4 but not M3 was detected (*Figure 4—figure supplement 3* and *Figure 4—figure supplement 5*). Also, the apparent labeling efficiency of M4 varied significantly

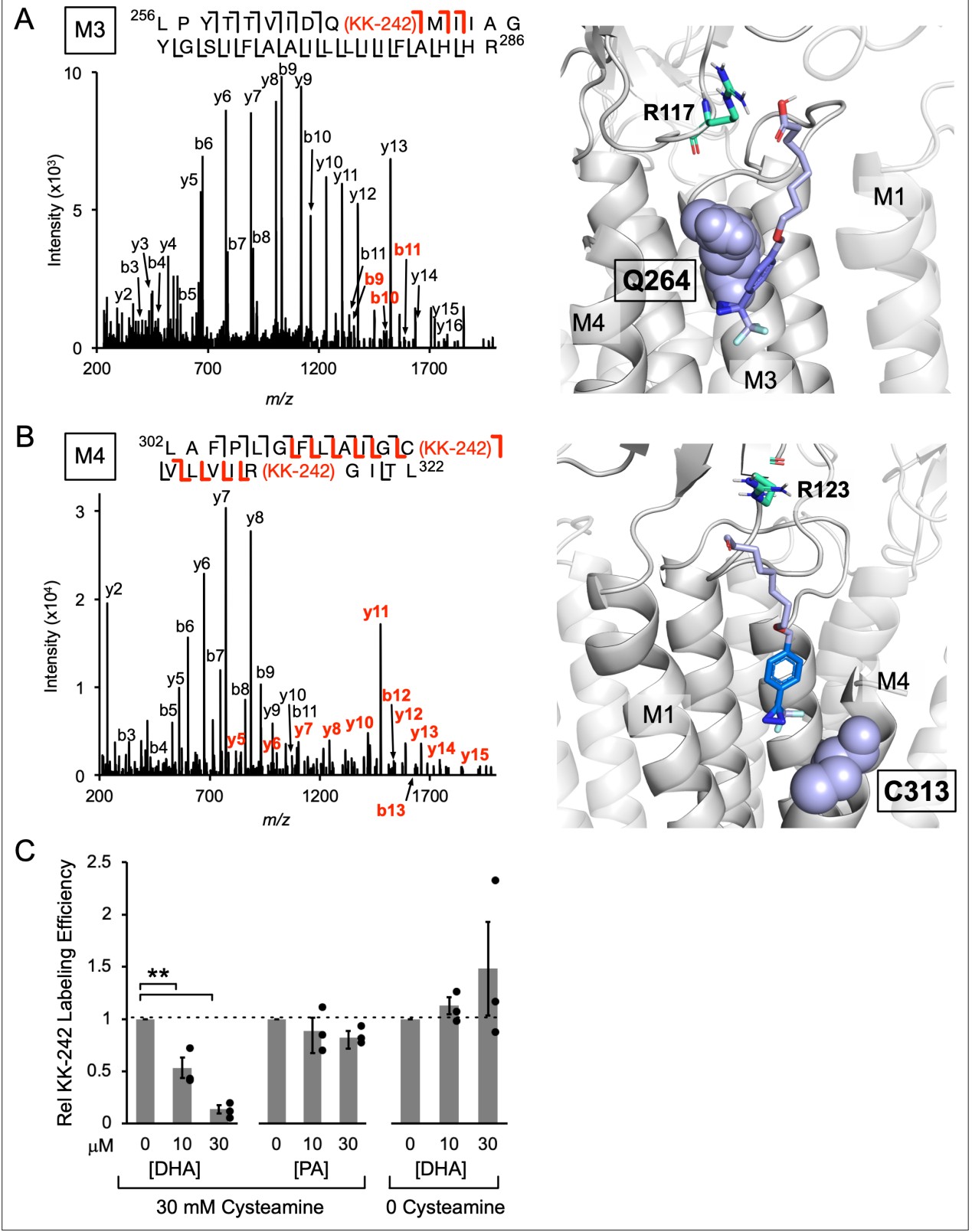

**Figure 4.** KK-242 photolabels two DHA binding sites in the outer TMD of ELIC. (**A**) *Left:* Fragmentation spectrum (MS2) of a M3 peptide photolabeled with KK-242 (*Figure 4—source data 1*). Red *b*-ions contain the KK-242 adduct mass, while black *b*- and *y*-ions do not. The photolabeled residue, Q264, is indicated with KK-242 in parenthesis. *Right:* ELIC structure showing a KK-242 binding pose from docking with the diazirine adjacent to Q264 (purple sphere). Also shown is R117 (green cyan stick). (**B**) *Left:* MS2 spectrum of a M4 peptide photolabeled with KK-242 (*Figure 4—source data 1*).

*Figure 4 continued on next page*

*Figure 4 continued*

The fragment ions indicate labeling at C313 and R318. *Right:* ELIC structure showing a KK-242 binding pose from docking with the diazirine adjacent to C313 (purple sphere). Also shown is R123 (green cyan stick). (**C**) Apparent labeling efficiencies of M4 by 10 μM KK-242 in the presence of 10 and 30 μM DHA and PA. For DHA, apparent labeling efficiencies were evaluated in the presence and absence of cysteamine. Apparent labeling efficiencies in the presence of DHA or PA were normalized to control within each replicate (n = 3,± SEM). Statistical analysis was performed on the unnormalized data (*Figure 4—source data 2*) using a linear mixed effects model (*Figure 4—figure supplement 5*). In the presence of 30 mM cysteamine, the effect of DHA on KK-242 labeling efficiency was statistically significant (p = 0.016). See *Figure 4—source data 3* for the full description and results of the analysis.

The online version of this article includes the following source data and figure supplement(s) for figure 4:

**Source data 1.** Source data for *Figure 4A and B*.

**Source data 2.** Source data for *Figure 4*.

**Source data 3.** Source data for *Figure 4*.

**Figure supplement 1.** Peptide coverage of ELIC from LC-MS/MS analysis.

**Figure supplement 2.** Extracted ion chromatograms of M4 and M3 unlabeled and labeled peptides.

**Figure supplement 3.** Apparent labeling efficiencies of M4 and M3 by KK-242.

**Figure supplement 4.** Molecular representations of KK-242 docked to two sites in ELIC.

**Figure supplement 5.** KK-242 photolabeling competition data with DHA and PA.

**Figure supplement 6.** DHA inhibition of ELIC in the presence of PA.

between replicates (*Figure 4—figure supplement 5*). This is likely because different reverse phase columns and emitter tips were used in the LC-MS/MS analysis for each replicate, which could differentially alter the ionization efficiency of unlabeled or labeled peptides. However, within a single replicate of samples, which used the same column and emitter tip, the apparent labeling efficiencies were reproducible and could be compared between samples (*Figure 4C* and *Figure 4—figure supplement 5*). A decrease in apparent labeling efficiency of M4 in the presence of DHA or PA would suggest that these fatty acids are competing for KK-242 at the photolabeled sites. While DHA caused a dose-dependent reduction in KK-242 apparent labeling efficiency of M4, PA caused a small, statistically insignificant reduction (*Figure 4C*). This result indicates that DHA binds to KK-242 photolabeled sites with higher affinity than PA and is consistent with the inhibitory effects of DHA and PA on ELIC and the CGMD results. While we do not know the relative contribution of each site to M4 photolabeling, the large decrease in M4 apparent labeling efficiency in the presence of DHA (~90% with 30 μM DHA) suggests that both photolabeled sites are occupied by DHA. Consistent with the specificity of these binding sites for DHA over PA, the potency of DHA inhibition of ELIC peak responses by giant liposome patch-clamping was not altered in the presence of 2 μM PA (*Figure 4—figure supplement 6*).

DHA inhibits ELIC peak responses but has minimal effect on the $EC_{50}$ of cysteamine activation; this suggests that DHA inhibits ELIC by stabilizing an agonist-bound non-conducting state (*Gielen and Corringer, 2018*; see Discussion). To test this possibility, we examined the effect of DHA on KK-242 apparent labeling efficiency of M4 in the absence of cysteamine. Indeed, DHA did not reduce KK-242 apparent labeling efficiency in the absence of cysteamine, in stark contrast to its effect in the presence of cysteamine (*Figure 4C*). Thus, DHA preferentially binds to KK-242 photolabeled sites when ELIC is in the agonist-bound state. It is worth noting that the apparent labeling efficiency of 10 μM KK-242 in the presence of cysteamine was only modestly higher than in the absence of cysteamine (*Figure 4—figure supplement 3*). Possible reasons for this include: (1) KK-242, being a weak inhibitor of ELIC, does not show the same state-dependence of binding as DHA, (2) the irreversible nature of photolabeling precludes detection of state-dependent photolabeling, and (3) apparent labeling efficiencies vary significantly between experiments due to factors mentioned above related to the LC-MS/MS. Nevertheless, the results clearly demonstrate that DHA preferentially binds to the photolabeled sites in an agonist-bound state.

Having identified two DHA binding sites on each face of M4 by photolabeling, we compared these results to the CGMD simulation data. We measured the contact probability of DHA with each amino acid in ELIC (*Figure 5*), to determine if DHA preferentially interacts with M1 or M3. While DHA has a non-zero contact probability with all lipid-facing transmembrane helices (i.e. M1, M3, and M4), residues on M3 have the highest cumulative contact probability (*Figure 5—figure supplement 1*). Interestingly, the residues with the highest contact probability between DHA and ELIC (A268, G271,

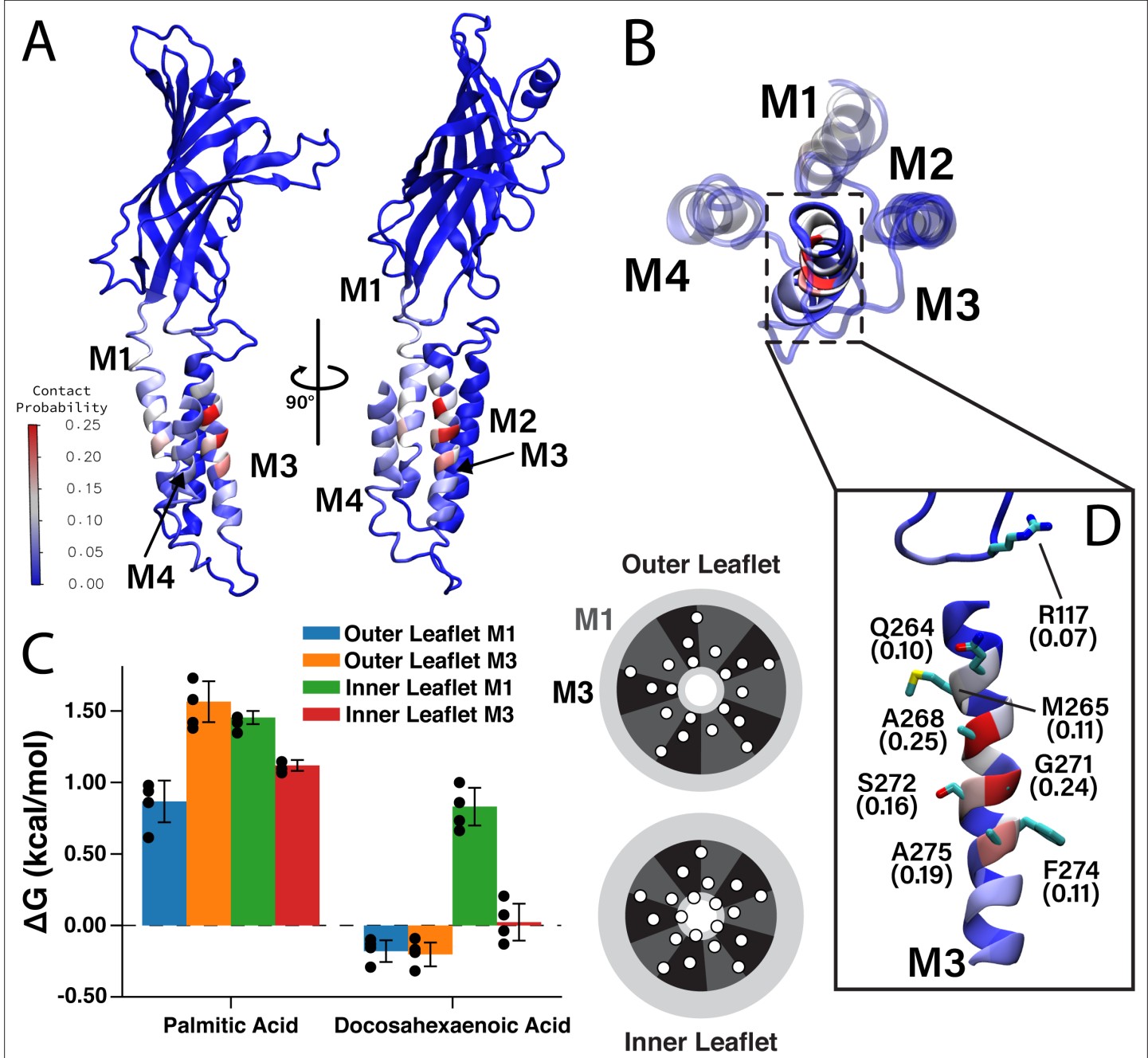

**Figure 5.** Favorable DHA binding sites from coarse-grained molecular dynamics. (**A**) An ELIC monomer is colored according to contact probabilities. The color bar to the left shows the contact probabilities values associated with the residue coloring. The same data is presented in (**B**) showing just the TMD as seen from the extracellular side. The M3 helix is shown in solid color, while M1, M2, and M4 are shown as transparent to highlight the DHA binding groove between M3 and M4. (**C**) Density threshold affinities for PA and DHA are shown (left) for the M1 and M3 sites in both the inner and outer leaflet (n = 4, ± SEM). To the right is a cartoon representation of the boundaries for the M1 (dark gray) and M3 (black) sites with the helices (white circles) representing the transmembrane domain. Each set of four same-colored helices represents one ELIC monomer. (**D**) The β6- β7 loop and M3 are colored according to contact probability. Residues in this binding site with the highest contact probabilities are highlighted by stick representation and the associated contact probabilities are shown.

The online version of this article includes the following figure supplement(s) for figure 5:

**Figure supplement 1.** Contact probability for PA (blue trace) and DHA (black trace) as a function of residue number.

**Figure supplement 2.** Representative binding mode of DHA in the M3/M4 site.

**Figure supplement 3.** Density threshold affinities of POPC, POPE, and POPG in the M1 and M3 sites from the CGMD simulations.

and A275) are located on a single face of M3 (*Figure 5D*), namely the face oriented toward M4, suggesting a specific DHA binding site in the groove between M3 and M4. R117 and Q264 (Q264 is photolabeled by KK-242) are located on the same face of M3 and interact with DHA, especially the carboxylate headgroup throughout the simulation (*Figure 5D*, *Figure 5—figure supplement 2*). The remaining residues on the binding face of M3 (A268, G271, S272, and A275) interact extensively with the polyunsaturated tail. These results reinforce the photolabeling data and identify the M3-M4 intra-subunit groove as a specific binding site for DHA. While M3 is noted to have the highest averaged contact probability (7.8% in M3 vs 5.3% in M1 and 4.7% in M4), a lower, but non-trivial, amount of contact is noted between DHA and M1 (*Figure 5A*, *Figure 5—figure supplement 1*), consistent with KK-242 photolabeling of C313.

To quantitatively compare the stability of DHA in each of these sites, we calculated a quantity analogous to a binding affinity, the density threshold affinity. The density threshold affinity ΔG (*Sharp and Brannigan, 2021*) represents the relative free-energy difference between a lipid species in the bulk membrane and in a specified, well-defined region near the protein, and is calculated using density ratios. Here, we define four sites per subunit, the 'M1 site' (dashed sectors in *Figure 2*) in each of the two leaflets, and the 'M3 site' (solid sectors in *Figure 2*) in each of the two leaflets. The site boundaries are inspired by the 2D density distributions in *Figure 2*, have fivefold symmetry, and encompass non-overlapping regions around each respective helix. The calculated values of ΔG for PA and DHA for each of the four sites are shown in *Figure 5C*. Consistent with radial enrichment results described earlier, DHA has higher affinity across all sites compared to PA. Indeed, PA has unfavorable interaction with all sites measured, having the lowest affinity for the outer leaflet M3 site (*Figure 5C*). Comparing affinity of DHA between different sites, there is a clear preference for outer leaflet sites versus inner leaflet sites, with both outer leaflet sites having favorable energy change and both inner leaflet sites having unfavorable energy change. This is consistent with the radial enhancement data and the fact that KK-242 was not observed to label inner leaflet sites. Although not statistically significant, there is slightly higher affinity of DHA for the outer membrane M3 site than the outer membrane M1 site (*Figure 5C*) in agreement with the contact probability analysis. Overall, the results from these analyses support the notion that DHA binds to outer leaflet intrasubunit sites.

## DHA inhibits ELIC through one high-affinity site

To investigate the functional significance of the M1 and M3 DHA binding sites, we modified each site with hexadecyl-methanethiosulfonate (hMTS); the hexadecyl group is identical to a PA alkyl tail. Although it does not mimic the polyunsaturated tail of DHA, we reasoned that covalent attachment of a hexadecyl group to these sites may produce an inhibitory effect, since PA is a weak inhibitor of ELIC. Based on the photolabeling, docking and CGMD results, we also reasoned that the fatty acid headgroup is near R117 and R123 when occupying either binding site (*Figures 4A, B and 6A*). Thus, we modified these positions with hMTS to mimic binding of a fatty acid. On a cysteine-less background (C300S/C313S mutant), we introduced R117C and R123C mutations. These triple mutants produced stable purified protein, and were incubated with 100 µM hMTS for 1 hr and analyzed by intact protein MS. Both mutants were completely modified by one hMTS per subunit consistent with specific modification of the introduced cysteine (*Figure 6A*). We previously showed that a cluster of three arginines in the inner interfacial region (R286, R299, R301) of the ELIC TMD form charged interactions with anionic phospholipids, which influence channel agonist responses (*Figure 6A*; *Tong et al., 2019*). Thus, we also generated cysteine mutations of these arginines; interestingly, these mutants were not modified by hMTS indicating decreased accessibility at this site (*Figure 6A*). This finding is consistent with CGMD results showing reduced binding of DHA and PA in the inner leaflet compared to the outer leaflet (*Figures 2 and 5C*). Moreover, analysis of the diacylated lipids species in the CGMD system (*Figure 5—figure supplement 3*) shows that POPG has a favorable affinity for this site, suggesting that higher occupancy of this site by POPG leads to the relative exclusion of fatty acids or hMTS.

Having modified R117C/C300S/C313S and R123C/C300S/C313S with hMTS (henceforth called R117C-M and R123C-M), we re-purified the modified proteins by size exclusion chromatography (SEC) to confirm a monodisperse sample and reconstituted in giant liposomes for excised patch-clamp recordings. While this approach worked for R123C-M, R117C-M in DDM was partially aggregated by SEC. To enable liposome reconstitution of R117C-M, we first equilibrated unmodified

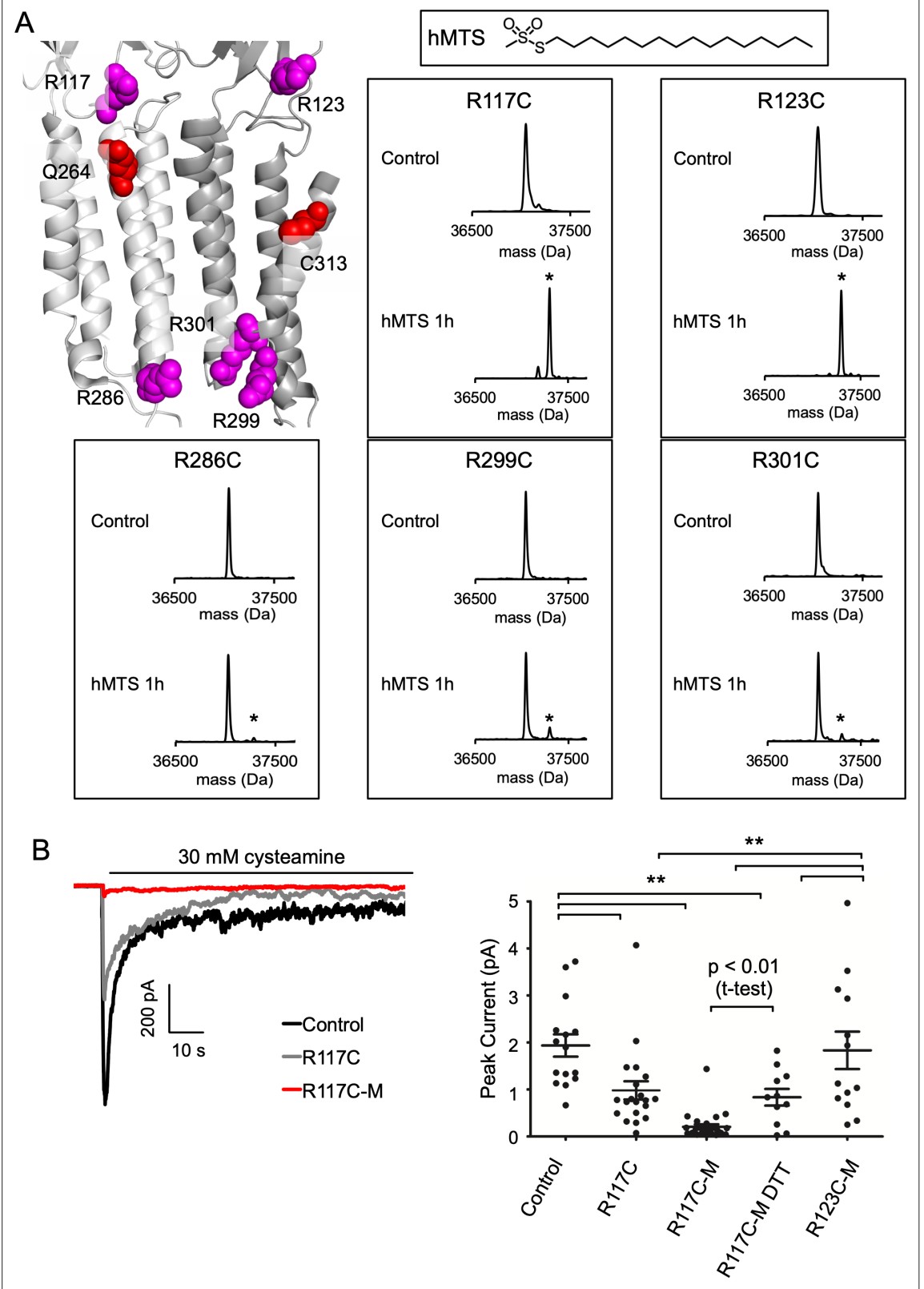

**Figure 6.** hMTS modification of R117C/C300S/C313S inhibits ELIC agonist response. (**A**) *Top left:* Structure of ELIC showing photolabeled residues (Q264 and C313), and arginines at the outer (R117, R123) and inner (R286, R299, R301) interfacial regions of the TMD. *Right:* Deconvoluted intact protein mass spectra of the indicated arginine to cysteine mutants on a C300S/C313S background. Mass spectra are shown prior to hMTS modification (control) and after 1 h incubation with hMTS. The peak in all control spectra corresponds to the mass of unmodified ELIC mutant (37,048 Da), and a + 257.5 Da

*Figure 6 continued on next page*

*Figure 6 continued*

shift indicates modification with one hMTS (marked by asterisk). (**B**) *Left:* Sample currents of control (C300S/C313S), R117C (C300S/C313S/R117C), and R117C-M (C300S/C313S/R117C modified with hMTS) from 2:1:1 POPC:POPE:POPG giant liposomes. *Right:* Average peak current from control, R117C, R117C-M, R117C-M treated with 2 mM DTT, and R123C-M in response to 30 mM cysteamine (n = 12–26,± SEM,). All conditions were compared using a one-way ANOVA and posthoc Tukey test (** p < 0.01). An unpaired T-test was also performed between R117C-M and R117C-M DTT (p < 0.01).

The online version of this article includes the following figure supplement(s) for figure 6:

**Figure supplement 1.** hMTS modification of R117C/C300S/C313S.

**Figure supplement 2.** Functional properties of ELIC cysteine mutants (hMTS modified or unmodified).

**Figure supplement 3.** Comparison of DHA binding in ELIC and GLIC.

**Figure supplement 4.** Simulated agonist responses demonstrating the effect of stabilizing an inhibitor (I)-bound pre-active state (AFI).

**Figure supplement 5.** Simulated agonist responses demonstrating the effect of stabilizing an inhibitor (I)-bound desensitized state (ADI).

R117C/C300S/C313S with DDM-destabilized liposomes, and then modified with hMTS such that R117C-M was maintained in a lipid environment (see Experimental Procedures). After this treatment, biobeads were added to complete liposome formation. Intact protein MS of R117C-M from this proteoliposome sample showed two main peaks corresponding to the modified mutant with and without a bound phospholipid (*Figure 6—figure supplement 1A*). This phospholipid adducted to a denatured ELIC subunit is likely due to the abundance of residual phospholipid in this sample. No unmodified mutant was detected; however, this spectrum had significantly more noise and adduction compared to ELIC samples in DDM. Based on this noise, we estimate that >90% of the R117C/C300S/C313S mutant was modified with hMTS but cannot rule out the presence of some unmodified protein. SDS-PAGE of protein from C300S/C313S and R117C-M proteoliposomes showed equal amounts of ELIC in both samples consistent with a successful reconstitution (*Figure 6—figure supplement 1B*).

Having produced liposomes with C300S/C313S (control), R117C-M or R123C-M, we examined channel responses to 30 mM cysteamine. Peak currents from excised patches of control and R123C-M were of similar magnitude, but peak currents of R117C-M were significantly smaller with many patches showing no current (*Figure 6B*). We also tested the R117C/C300S/C313S mutant (i.e. unmodified, labeled as R117C in *Figure 6B*) and found that this mutation significantly reduced peak currents compared to control; however, R117C-M peak currents were still lower than R117C/C300S/C313S (*Figure 6B*, not a significant difference by one-way ANOVA). To further verify that hMTS modification of R117C inhibits channel responses, we performed recordings from R117C-M proteoliposomes incubated in 2 mM DTT, which may remove the disulfide-mediated hMTS modification. R117C-M peak currents in the presence of 2 mM DTT were significantly larger than in the absence of DTT (p < 0.01 by unpaired T-test; note that R117C-M and R117C-M DTT currents were recorded from the same batch of liposomes) (*Figure 6B*). While there is substantial variability in peak currents between patches, the results suggest that hMTS modification of R117C inhibits channel responses, mimicking the effect of fatty acids.

The C300S/C313S mutant (control) produced slower current decay and larger steady state currents compared to WT (*Figure 1—figure supplement 1* and *Figure 6—figure supplement 2*), which is consistent with previous reports of the effect of mutations at these residues (*Kinde et al., 2015*). However, cysteamine responses for R117C/C300S/C313S, R117C-M and R123C-M did not show significantly different rates of current decay or relative steady state current compared to control (*Figure 6—figure supplement 2*). Moreover, all four showed similar relative reduction of peak current with pre-application of 30 µM DHA (*Figure 6—figure supplement 2*, see discussion for commentary on why currents from R117C-M liposomes remain sensitive to DHA).

## Discussion

Using a novel fatty acid photolabeling reagent and CGMD, we identified two fatty acid binding sites in ELIC, which are occupied by the PUFA, DHA, with higher affinity than the saturated fatty acid, PA. The striking agreement between photolabeling and simulation results provides a compelling molecular description of the selectivity of fatty acid interactions with a pLGIC. Along with site-directed hMTS modification to interrogate the functional significance of these sites, the results strongly suggest that DHA inhibits ELIC through a single site, an intrasubunit groove between the top of M3 and M4. This

site is analogous to the DHA binding site in a GLIC crystal structure (*Basak et al., 2017*). That this fatty acid binding site is present in both ELIC and GLIC suggests that it may also be conserved in eukaryotic pLGICs. Indeed, the equivalent site in the β3 subunit of the α1β3 GABA$_A$R was found to be a neurosteroid binding site that mediates neurosteroid inhibition of GABA$_A$R responses (*Sugasawa et al., 2020*). Thus, this site may be a conserved lipid binding site by which fatty acids and neurosteroids inhibit pLGICs.

Compared to the GLIC crystal structure, our photolabeling and CGMD results provide additional and contrasting information on DHA binding at the M3/M4 site. We illustrate this in *Figure 6—figure supplement 3* by aligning the GLIC-DHA crystal structure with the color-coded DHA contact probabilities in ELIC from the CGMD results in *Figure 5A*. In the GLIC structure, the DHA carboxylate headgroup interacts with R118 in the β6-β7 loop, but the polyunsaturated tail is not interacting significantly with the TMD and is mostly unresolved (*Figure 6—figure supplement 3*). In contrast, we find that the polyunsaturated tail of DHA, more so than the headgroup, strongly interacts with the M3/M4 groove of ELIC making high probability contacts with small hydrophobic residues such as A268 and G271 (*Figure 5A* and *Figure 6—figure supplement 3*). Moreover, DHA binds to this site with much higher affinity than PA. Since both DHA and PA have an anionic carboxylate headgroup, this indicates that the polyunsaturated tail is the primary determinant of DHA binding affinity. Consistent with the nature of DHA binding to the M3/M4 site, we find that:(1) DHA methyl ester inhibits ELIC with similar potency and efficacy as DHA, and(2) DHA inhibits R117C/C300S/C314S and C300S/C314S similarly. Thus, a charged interaction between the carboxylate headgroup of DHA and R117 (equivalent to R118 in GLIC) (*Figure 6—figure supplement 3*) is not critical for DHA binding and effect in ELIC. Both DHA and DHA methyl ester modulate hippocampal excitability through the GABA$_A$R (*Taha et al., 2013*) suggesting that this mechanism of PUFA inhibition is conserved in mammalian pLGICs. The discrepancy between our findings and the GLIC structure may be a consequence of different methodologies or a difference in DHA interactions between ELIC and GLIC.

DHA binding to KK-242 photolabeled sites was specific for the agonist-bound state. This result supports a direct binding mechanism in which DHA allosterically inhibits ELIC responses by binding to and stabilizing an agonist-bound non-conducting state. Our patch-clamp data show that fatty acids inhibit ELIC peak responses when applied prior to agonist similar to previous reports in the GABA$_A$R and nAchR. This is in contrast to the effect of DHA described in GLIC, in which DHA was co-applied with agonist resulting in no change in peak responses but faster and greater current decay (*Basak et al., 2017*). This apparent increase in the rate and extent of current decay was interpreted as fatty acid stabilization of a desensitized state. However, a different hypothesis was recently proposed by Gielen and Corringer for understanding the(1) increase in desensitization rate with fatty acid co-application and(2) decrease in peak response with fatty acid pre-application; both of these findings can be predicted by a model in which a pre-active state is stabilized (*Gielen and Corringer, 2018*). Our results with ELIC show that pre-application of DHA not only inhibits peak current, but also results in faster and more profound current decay. We have repeated the simulations performed by Gielen and Corringer using the same gating model (*Gielen and Corringer, 2018*), and found that the qualitative effects of fatty acid inhibition observed in ELIC (i.e. both a decrease in peak current and increase in the rate/extent of current decay with fatty acid pre-application) can be produced by stabilizing either a pre-active (*Figure 6—figure supplement 4*) or desensitized state (*Figure 6—figure supplement 5*). The model predicts that stabilization of a pre-active or desensitized state, to the extent that peak responses are completed inhibited, will lead to most channels occupying these respective states at steady-state (*Figure 6—figure supplement 4C* and *Figure 6—figure supplement 5C*). Thus, more structures of pLGICs with fatty acids in the presence of agonist may reveal whether fatty acids stabilize pre-active or desensitized conformations of the channel.

By introducing a hexadecyl group to fatty acid binding sites using hMTS, we sought to mimic occupancy of these sites by PA. Alkyl-MTS modification has been previously used to examine the effects of phospholipids at specific sites in inward rectifying potassium channels (*Lee et al., 2013*; *Enkvetchakul et al., 2007*). In this study, we combined hMTS modification with intact protein MS to measure modification efficiency. This proved useful in showing that hMTS has limited access to a cluster of arginine residues in the inner TMD of ELIC, shown previously to be a binding site for the anionic phospholipid, phosphatidylglycerol (PG) (*Hénault et al., 2019*; *Tong et al., 2019*; *Sridhar et al., 2021*). Since ELIC in DDM co-purifies with PG from *E. coli* membranes (*Tong et al., 2019*), it is

plausible that hMTS cannot access these sites due to the presence of tightly bound PG. KK-242 may have not photolabeled sites in the inner TMD for the same reason. Indeed, the coarse-grained simulations with membranes containing 25% POPG showed significant reduction in the binding of DHA to the inner TMD of ELIC due to enrichment of POPG at this site. It is interesting to note that fatty acids are thought to modulate Kv channels by interacting with positively charged residues also in the extracellular side of the voltage-sensor domain (*Börjesson et al., 2008*; *Farag et al., 2016*; *Yazdi et al., 2021*). Thus, fatty acids may be excluded from sites in the inner TMD of many ion channels due to the presence of anionic phospholipids, which are localized in the inner leaflet of the cell membrane and bind with relatively high affinity to sites in the inner TMD (*Lee et al., 2013*; *Cheng et al., 2011*; *Hite et al., 2014*; *Gao et al., 2016*; *Hansen et al., 2011*). Likewise, lipids in the outer leaflet may compete for binding of fatty acids to the M3/M4 groove. Indeed, a recent structure of ELIC in SMA nanodiscs extracted from *E. coli* membranes showed cardiolipin bound to approximately this site (*Kumar et al., 2021*). In another study, MD simulations suggest that zwitterionic phospholipids such as PC occupy this site (*Sridhar et al., 2021*). It is plausible that cardiolipin, or other lipids in mammalian pLGICs, could compete for DHA binding to this site and alter the potency of DHA inhibition.

The strong inhibitory effect of alkylation of R117C is similar to the maximal effect of DHA or DHA methyl ester, but different from PA which has a much weaker effect. A likely explanation for this discrepancy is that PA has lower occupancy of this site compared to DHA or a covalently attached hMTS, resulting in a lower apparent efficacy. Indeed, PA was much less effective in reducing KK-242 photolabeling compared to DHA and was depleted around ELIC in the coarse-grained simulations. Thus, the difference in inhibition between DHA and PA appears to be entirely related to different binding affinities for this site. If a hexadecyl group resembling PA is covalently introduced to this site (i.e. 100% occupancy of this site), it appears as effective as DHA at inhibiting ELIC responses.

If hMTS modification of R117C occupies a DHA binding site, it may occlude the inhibitory effect of DHA; this was not the case with R117C-M. Since the small currents observed in R117C-M (~10% of control) remained sensitive to DHA, it is possible that these currents arise from ~10% of R117C/C300S/C313S that was not modified. This unmodified protein may have been missed by intact protein MS due to the high noise in the spectrum. We also cannot exclude the possibilities that: (1) DHA can still partially displace a covalently attached hMTS from this site, or (2) DHA also acts through an entirely different mechanism. Regardless of what accounts for these small DHA-sensitive currents, the strong inhibitory effect of R117C-M and the reversal of this effect by DTT support the hypothesis that this site mediates fatty acid inhibition of ELIC.

The photolabeling results with ELIC indicate that KK-242 is a useful fatty acid analogue photolabeling reagent. KK-242, which features a TPD, is advantageous compared to pacFA, which features an aliphatic diazirine, for the identification of fatty acid binding proteins and binding sites. This is likely because the TPD can label any amino acid including aliphatic and aromatic side chains, which usually line the binding pockets of fatty acid alkyl tails (*Simard et al., 2005*). This limitation of pacFA (i.e. preferential labeling of nucleophilic residues by aliphatic diazirines) likely applies to other lipid analogue photolabeling reagents that feature an aliphatic diazirine in the hydrophobic regions of the molecule (e.g. PhotoClick-Cholesterol, pacFA ceramide) potentially reducing the utility of these reagents for identifying lipid binding proteins (*Haberkant et al., 2013*; *Dadsena et al., 2019*; *Hulce et al., 2013*) or lipid binding sites.

In conclusion, DHA inhibits ELIC through specific binding to an M3/M4 intrasubunit site in the agonist-bound state.

## Materials and methods

### Key resources table

| Reagent type (species) or resource | Designation | Source or reference | Identifiers | Additional information |
|---|---|---|---|---|
| Strain, strain background (*Escherichia coli*) | BL21(DE3) C43 | Lucigen | Catalogue # 60446–1 | electrocompetent cells |
| Recombinant DNA reagent | Pet26-MBP-ELIC | Addgene | #39,239 | ELIC plasmid |

*Continued on next page*

*Continued*

| Reagent type (species) or resource | Designation | Source or reference | Identifiers | Additional information |
|---|---|---|---|---|
| Commercial assay or kit | QuikChange XL | Agilent Technologies | Catalogue # 200,516 | Site-Directed Mutagenesis Kit |
| Chemical compound, drug | cis-4,7,10,13,16,19-Docosahexaenoic acid | Millipore Sigma | D2534 | polyunsaturated fatty acid |
| Chemical compound, drug | Cysteamine | Millipore Sigma | M9768 | ELIC agonist |
| Chemical compound, drug | Hexadecyl methanethiosulfonate | Santa Cruz Biotechnology | sc-215144 | Alkyl-MTS reagent |
| Peptide, recombinant protein | Trypsin from porcine pancreas | Millipore Sigma | T6567 | |
| Other | PLRP-S | Agilent | PL1114-9999 | Reverse-phase column |
| Software, algorithm | XCalibur2.2 | Thermo Scientific | | MS analysis |
| Software, algorithm | PEAKS X+ (10.5) | Bioinformatics Solutions Inc | | MS analysis |
| Software, algorithm | pClamp10.4 | Molecular Devices | RRID:SCR_011323 | Patch-clamp software |
| Software, algorithm | R | https://www.r-project.org/ | | Statistical computing |

## Expression, purification, and mutagenesis of ELIC

ELIC was expressed and purified as previously described (*Tong et al., 2019*), yielding purified ELIC in 10 mM Tris pH 7.5, 100 mM NaCl, and 0.02% DDM (Buffer A). Site-directed mutagenesis was performed by a standard QuikChange approach and all mutations were confirmed by sequencing (Genewiz).

## ELIC giant liposome formation and excised patch-clamp recordings

ELIC WT or mutants were reconstituted in giant liposomes as previously described (*Tong et al., 2019*). For each liposome preparation, 0.5 mg of ELIC protein was reconstituted in 5 mg of 2:1:1 POPC:POPE:POPG liposomes, and the entire procedure as well as patch-clamp recordings was performed using 10 mM MOPS pH 7 and 150 mM NaCl (MOPS buffer). Giant liposomes were formed by dehydrating 10 μl of proteoliposomes on a glass coverslip in a desiccator for 1 hr at RT followed by rehydration with 100 μl MOPS buffer overnight at 4 °C and 2 hr at RT the following day. Giant liposomes were suspended by pipetting and then applied to the recording bath in MOPS buffer.

Excised patch-clamp recordings of ELIC in giant liposomes were performed as previously described (*Tong et al., 2019*). Bath and pipette solution contained MOPS buffer and 0.5 mM BaCl₂. Fatty-acid-containing solutions were prepared from DMSO stocks of 30 mM for each fatty acid. Thus, the final concentration of DMSO was 0.1% or less in all fatty-acid-containing solutions, and 0.1% DMSO was added to control solutions to maintain consistency. Treatment of R117C-M proteoliposomes with DTT was performed by incubating the proteoliposomes in 2 mM DTT, and including 2 mM DTT in control and 30 mM cysteamine solutions. All recordings were voltage-clamped at –60 mV and data collected at 5 kHz using an Axopatch 200B amplifier and Digidata 1440 A (Molecular Devices, San Jose, CA) with Axopatch software. Rapid solutions changes were performed using a three-barreled flowpipe mounted to and controlled by a SF-77B fast perfusion system (Warner Instrument Corporation, Hamden, CT). Using changes in liquid junction current with an open pipette, the 10–90% exchange time was determined to be ~2 ms. Current decay was fit with a double exponential, and weighted time constants were obtained using:

$$Weighted\,Tau = \frac{(A1 \cdot \tau 1) + (A2 \cdot \tau 2)}{A1 + A2}$$

(1)

where A1 and A2 are the amplitudes of the first and second exponential components. Peak responses as a function of cysteamine were fit to a Hill equation with n = 2 to derive an $EC_{50}$. Statistical analysis was performed using multiple comparisons with a Dunnett's test to compare fatty acid or KK-242 inhibition with control.

## Photolabeling of ELIC and intact protein MS

Photolabeling of ELIC by pacFA and KK-242 and intact protein MS analysis was performed as previously described (*Cheng et al., 2018*). Fifty µg of purified ELIC in buffer A was mixed with 100 µM pacFA or KK-242 for 30 min, and irradiated with >320 nm UV light for 5 min. For intact-protein MS analysis, 25 µg of photolabeled ELIC was precipitated with chloroform, methanol and water, pelleted and reconstituted in 3 µl of 100% formic acid for 10 s followed by 50 µl of 4:4:1 chloroform/methanol/water. These samples were analyzed in the ion trap (LTQ) of an Elite mass spectrometer (Thermo Scientific) by direct injection using a Max Ion API source with a HESI-II probe. Data was acquired with a flow rate of 3 µl/min, spray voltage of 4 kV, capillary temperature of 350 °C, and SID of 30 V. Spectra were deconvoluted using Unidec (*Marty et al., 2015*).

## Middle-down MS analysis of photolabeled ELIC

To sequence photolabeled ELIC, a tryptic middle-down MS analysis was performed as previously described with some modifications (*Cheng et al., 2018*). Fifteen µg of KK-242 photolabeled ELIC was buffer exchanged to 50 mM triethylammonium bicarbonate (TEABC) pH 7.5% and 0.02% DDM using Biospin gel filtration spin columns (Bio-Rad). These samples were then reduced and alkylated with triscarboxyethylphosphine (TCEP) and N-ethylmaleimide (NEM) followed by digestion with 2 µg of trypsin for 7 days at 4 °C. Digestion was terminated with formic acid at 1%. The final sample volume was 100 µl, and 20 µl was analyzed by LC-MS using an in-house PLRP-S (Agilent) column and Orbitrap Elite mass spectrometer (Thermo Scientific). MS/MS was performed as previously reported (*Sugasawa et al., 2020*), using HCD (higher energy collisional dissociation). The LC-MS data was searched with PEAKS (Bioinformatics Solutions) using the following search parameters: precursor mass accuracy of 20 ppm, fragment ion accuracy of 0.1 Da, up to three missed cleavages on either end of the peptide, false discovery rate of 0.1%, and variable modifications of methionine oxidation, cysteine alkylation with NEM, and the KK-242 mass (330.14 Da). Manual data analysis of MS1 and MS2 spectra was performed with XCalibur (Thermo Scientific). Photolabeled peptides were only accepted if there was a corresponding unlabeled peptide with a shorter a retention time, and a mass accuracy of <10 ppm. All fragment ions for MS2 spectra had a mass accuracy of <20 ppm.

To assess competition of KK-242 photolabeling by fatty acids, 15 µg of ELIC was photolabeled with 10 µM KK-242 in the absence and presence of different concentrations of DHA and PA. The apparent labeling efficiency of M4 was obtained by taking the area under the curve of extracted ion chromatograms of unlabeled and labeled M4 in XCalibur. Apparent labeling efficiency was calculated as the abundance (area under the curve) of labeled peptide/(unlabeled+ labeled peptide).

hMTS modification of ELIC mutants and reconstitution in giant liposomes hMTS modification of ELIC was performed by mixing ELIC in buffer A at 1 mg/ml with equal volumes of 200 µM hMTS in buffer A for a final concentration of 100 µM hMTS and 0.5 mg/ml ELIC. This sample was incubated at RT for 30 min, followed by removal of 25 µg of protein for intact protein MS analysis and purification of the remaining protein on a Sephadex 200 Increase 10/300 column. For hMTS-modified R123C/C300S/C313S, this yielded a monodisperse protein, and 0.5 mg was reconstituted in giant liposomes in the same way as WT or C300S/C313S. Giant liposomes of hMTS-modified R117C/C300S/C313S were prepared as follows. 0.5 mg of purified protein was added to DDM-destabilized liposomes at 10 mg/ml (sample volume of 0.5 ml). After equilibrating for 30 min at RT, 0.5 ml of 200 µM hMTS in buffer A was added three times with each addition separated by 15 min (final volume 2 ml). Next, Bio-Beads were added to the remaining sample to remove DDM, as per the giant liposome formation protocol. SDS-PAGE analysis of ELIC protein from proteoliposomes was performed by solubilizing 2 µl of proteoliposomes (from frozen aliquots) in 1% SDS for 1 hr at RT. Samples were then run on 10% Tris-glycine polyacrylamide gel (Bio-Rad), and protein was detected using SYPRO-Ruby staining (Thermo Fisher) and fluorescence imaging of the gel.

## Docking of KK-242 to photolabeled sites in ELIC

Docking was performed using Autodock 4.2 as previously described (*Cheng et al., 2018*; *Morris et al., 2009*). The structure of KK-242 was generated in Maestro (Schrödinger), and Gasteiger charges and free torsion angles were determined by Autodock Tools. 2YN6 was used as the docking template, and two runs were performed with grids encompassing the two KK-242-photolabeled sites. The grid dimensions were 26 × 16 x 20 Å with 1 Å spacing for both sites and were centered to encompass the

outer TMD interface and C313 or Q264 for each site. Each docking run produced 300 poses that were clustered at 5.0 Å RMSD. Poses selected for *Figure 4* were taken from a cluster where the diazirine is closest to the photolabeled residue.

## Coarse-grained simulations of ELIC with fatty acid

For coarse-grained (CG) molecular dynamics simulations, the propylamine-bound non-conducting structure of ELIC (PDB 6V03) was used as the structural model (*Kumar et al., 2020*). As the deposited PDB terminates at residue 317, residues 318–321 (318-RGIT-321) were modeled assuming M4 retained an α-helical nature until termination. The pKa of each sidechain was measured using PROPKA (*Olsson et al., 2011*) and protonated assuming a pH of 7. The resulting ELIC structure was embedded and oriented in a POPC membrane using the CHARMM-GUI application (*Jo et al., 2008*; *Wu et al., 2014*). CHARMM-GUI was also utilized to solvate and ionize the system with 150 mM NaCl. The final system contained approximately 192,000 atoms and measured $122 \times 122 \times 143$ Å³. To allow the additional residues on the M4 helix to relax, an equilibration of the all-atom system was performed. The simulation system was energy minimized for 10,000 steps. While keeping the protein backbone harmonically restrained ($k = 5$ kcal•mol⁻¹•Å⁻²), the system was equilibrated for 5.0 ns in an NPT ensemble, allowing the membrane and aqueous phase to relax. The harmonic restraints on the protein backbone were then gradually released over a period of 5.0 ns under an NPT ensemble, after which the whole system was equilibrated without restraints for 5.0 ns. All atomistic simulations were performed with NAMD 2.14 (*Phillips et al., 2020*) with CHARMM36 parameters (*Klauda et al., 2010*; *Huang et al., 2017*). Temperature was maintained at 310 K using a Langevin damping coefficient, y, of 1 ps⁻¹. Pressure was maintained at 1.0 atm using the Nosé-Hoover Langevin piston method. Long-range electrostatic interactions were treated with particle mesh Ewald sums on a grid density >1 Å⁻³. A timestep of 2.0 fs was used for all atomistic simulations with bonded and non-bonded forces calculated every timestep and particle mesh Ewald sums calculated every other timestep. Long-range non-bonded interactions were cutoff after 12.0 Å with a smoothing function applied between 10.0 Å and 12.0 Å.

The resulting equilibrated ELIC structure was coarse-grained using the *martinize* script (*de Jong et al., 2013*), including secondary structural restraints. Conformational structure was maintained via harmonic restraints for backbone beads < 0.5 nm apart using a force constant of 1000 kJ mol⁻¹ with backbone pairs generated using the ElNeDyn algorithm (*Periole et al., 2009*). The coarse-grained ELIC structure was embedded in the membrane using the *insane* script (*Wassenaar et al., 2015*). Each simulation system used a bulk membrane composed of 2:1:1 POPC:POPE:POPG to which either 4 mol% palmitic acid or 4 mol% docosahexaenoic acid was added. For each system simulated, this equated to 15 fatty acid molecules per leaflet, or 30 total for the system. After solvation and ionization to 150 mM NaCl, final system size was approximately $16 \times 16 \times 20$ nm³ with approximately 44,000 beads.

Two initial simulation systems, one containing PA and one containing DHA, were created and equilibrated as below. Each system was energy minimized using steepest descent for approximately 30,000 steps and then equilibrated for 6.0 ns. The system was equilibrated in a NVT ensemble for 1.0 ns using the Berendsen thermostat set at 323 K and a temperature coupling constant of 1.0 ps. Following this, the system was equilibrated in a NPT ensemble for 5.0 ns using the Berendsen thermostat and barostat. The temperature was set at 323 K with a temperature coupling constant of 1.0 ps; the pressure was maintained at 1.0 bar using a pressure coupling constant of 3.0 ps and compressibility of $3.0 \times 10^{-5}$ bar⁻¹.

Once the initial systems were equilibrated, four replicates of each experimental condition were simulated for 10 μs, meaning a total of 40 μs of simulation data for PA and DHA each. For production simulations, temperature was maintained at 323 K using the velocity rescaling algorithm (*Bussi et al., 2007*) with the coupling constant set to 1.0 ps. The protein and lipid beads were coupled to a temperature bath separate from water and ions. Each bath had the same temperature parameters. Pressure was maintained at 1.0 bar semi-isotropically utilizing a time constant of 12 ps and a compressibility of $3.0 \times 10^{-5}$ bar⁻¹ with the Parinello-Rahman coupling scheme. All coarse-grained simulations were performed with GROMACS 2020.4 and the MARTINI 2.2 force field (*de Jong et al., 2013*; *Wassenaar et al., 2015*). Electrostatics were treated with a reaction-field and dielectric constant of $\varepsilon = 15$; van der Waals and Coulomb interactions were cut-off after 1.1 nm. A timestep of 25 fs was used throughout. Frames were saved every 1.0 ns and used for analysis as described below.

To understand the boundary distribution of fatty acids, the boundary lipid metric, B, was measured according to:

$$B_i = \left\langle \frac{b_i}{b_{tot}} \right\rangle \chi_i^{-1} \qquad (2)$$

where $b_i$ is the number of lipids of species $i$ in the boundary shell, $b_{tot}$ is the total number of lipids in the boundary shell, and $\chi_i$ is the fraction of lipid species $i$ in the model membrane. The brackets denote an average over the time of the simulation. A lipid was considered in the boundary shell if it was within 6 Å of the protein.

Two-dimensional radial enhancement calculations were performed as demonstrated previously (*Tong et al., 2019*; *Sharp et al., 2019*; *Sharp and Brannigan, 2021*) to characterize fatty acid interactions with ELIC. The two-dimensional radial density distribution, $\rho_B$, was calculated according to:

$$\rho_B = \frac{\langle n_B(r_i, \theta_j) \rangle}{r_i \Delta r \Delta \theta} \qquad (3)$$

where ‹$n_B(r_i, \theta_j)$› is the time-averaged number of beads of lipid species $B$ in the bin centered around radius $r_i$ and polar angle $\theta_j$, $\Delta r$ is the radial bin width (5 Å in this study), and $\Delta \theta$ is the polar angle bin width ($\pi/15$ radians in this study). In order to determine relative enrichment or depletion of the lipid compared to the bulk membrane phase, the two-dimensional radial density distribution was normalized to expected bead density of lipid species $B$ in the bulk:

$$\widetilde{\rho}_B = \frac{\rho_B}{x_B s_B N_L \langle A \rangle^{-1}} \qquad (4)$$

where $x_B$ is the mol fraction of lipid species $B$, $s_B$ is the number of coarse-grained beads in lipid species $B$, $N_L$ is the total number of lipids in the simulation system, and ‹$A$› is the average projected area of the simulation box in the plane of the membrane.

Contact probabilities of each individual residue with fatty acids were determined as follows. In each simulation frame, if any fatty acid bead was located within 5 Å of any bead of the residue, a contact was counted for that frame. The number of contacts was then time-averaged over the simulation and then averaged across all five subunits.

A density affinity free-energy analysis was performed as previously described (*Sharp and Brannigan, 2021*). Briefly, this analysis determines the relative free-energy difference for a lipid species between the bulk membrane and a particular binding site by comparing the probability distribution of finding $n$ coarse-grained fatty acid beads in the binding site, $P_{site}(n)$, versus the same probability distribution for bulk membrane, $P_{bulk}(n)$. In this study, we defined two new binding sites, termed 'M1' and 'M3' (*Figures 2 and 5C*). For the M1 site, one angular boundary was determined to be the M4 helix and the other angular boundary to be midway between the M1 helix of the same subunit and the M3 helix of the adjacent subunit. For the M3 site, one angular boundary was determined to be the M4 helix and the other angular boundary to be midway between the M3 helix of the same subunit and the M1 helix of the adjacent subunit. The radial boundaries for both sites were $10 < r < 40$ Å. $P_{site}(n)$ was determined by counting the number of coarse-grained fatty acid beads within the boundaries of each binding site described above (i.e. M1 and M3) for each frame and normalizing to the number of frames in the simulation. As $P_{bulk}(n)$ needs to be corrected for the area of $P_{site}(n)$, a separate simulation was conducted to determine the area of the M1 and M3 sites (*Sharp and Brannigan, 2021*). Exploiting the relationship that $P_{site}(n)$ is given for each individual CGMD simulation run underneath the two-dimension = $P_{bulk}(n)$ in a unitary membrane, a coarse-grained system containing ELIC (PDB 6V03) in a POPC membrane was simulated under the same conditions described above for 10 µs.

## Macroscopic current simulations with Channelab

Simulations of current responses were performed using Channelab utilizing a pLGIC gating model as previously described (*Gielen and Corringer, 2018*). Currents were simulated using the model and rate constants indicated in *Figure 6—figure supplements 4 and 5*, and a fourth-order Runge-Kutta integration algorithm. Each simulation was run pre-equilibrated in the presence of varying concentrations of inhibitor, followed by the application of agonist for 25 s. Channel currents were determined by monitoring the probability of occupying the AO or AOI states. Occupancy of pre-active or

desensitized states were monitored by calculating the probability of occupying the AF/AFI or AD/ADI states, respectively.

## Statistical analysis

The photolabeling competition data in *Figure 4—figure supplement 5* were analyzed using a linear mixed effects model in the statistical computing program, R. This was performed using the lme4 package, and by setting DHA, PA and cysteamine as fixed effects.

## Synthesis of KK-242

**Scheme 1.** Synthesis of KK-242.

## 8-(*tert*-Butyl-dimethyl-silanyoxy-1-ol) (2)

1,8-Octanediol (1,730 mg, 5 mmol), imidazole (408 mg, 6 mmol) and *t*-butyldimethylsilyl chloride (750 mg, 5 mmol) were stirred in DMF (8 mL) for 15 hr. Saturated aqueous $NH_4Cl$ was added to the reaction and the product was extracted into ethyl acetate. The combined ethyl acetate extracts were washed with brine, dried over anhydrous $Na_2SO_4$, filtered and the solvent removed under reduced pressure on a rotary evaporator. Flash column chromatography on silica gel yielded purified product 2 (630 mg, 48.4%). $^1H$ NMR (400 MHz, $CDCl_3$) δ 3.68–3.54 (m, 4 H), 1.65–1.20 (m, 12 H), 0.90 (s, 9 H), 0.05 (s, 6 H).

## 3-[4-(8-*tert*-Butyl-dimethyl-silanyloxy)-octyloxymethylphenyl]-3-trifluoromethyl-3*H*-diazirine (3)

Sodium hydride in mineral oil (400 mg, 10 mmol), 3-(4-iodomethyl-phenyl)–3-trifluoromethyl-3*H*-diazi rine (300 mg, 0.92 mmol) and compound 2 (200 mg, 0.77 mmol) in THF (15 mL) were heated at reflux for 2 hr (it is critical to stop the reaction after 2 hours to prevent product decomposition). The reaction mixture was cooled to 0 °C and the excess sodium hydride were carefully quenched by adding cold water. Additional water (50 mL) was added and the product was extracted into ethyl acetate (40 mL x 3). The combined organic extracts were washed with brine, dried over anhydrous $Na_2SO_4$, filtered and the solvent removed under reduced pressure on a rotary evaporator to give the compound three as a yellow oil which was passed through a short column of silica gel to remove less polar impurities. The column was first eluted with hexanes followed by 5% ethyl acetate in hexanes to give the product **3**, which was sufficiently pure to be converted to compound **4**.

## 8-[4-(3-Trifluoromethyl-3*H*-diazirin-3-yl)benzyloxy-octan-1-ol] (4)

Partially purified compound four was dissolved in methanol (5 mL) and a freshly prepared 5% dry-HCl-methanol solution made by adding acetyl chloride to methanol (3 mL) was added and the reaction was stirred for 1 hr. The reaction was made basic by adding aqueous saturated $NaHCO_3$, and the product was extracted into dichloromethane (40 mL x 3). The combined organic extracts were washed with brine, dried over anhydrous $Na_2SO_4$, filtered and the solvent removed under reduced pressure on a rotary evaporator to give an oil which was purified by column chromatography (silica gel eluted with 15–25% ethyl acetate in hexanes) to give compound **4** (90 mg, 34%). [1]H NMR (400 MHz, $CDCl_3$) δ 7.37 (d, 2 H, *J* = 7.2 Hz), 7.18 (d, 2 H, *J* = 7.1 Hz), 4.50 (s, 2 H), 3.63 (t, 2 H, *J* = 6.2 Hz), 3.46 (t, 2 H, *J* = 6.2 Hz), 1.70–1.20 (m, 12 H). [13]C NMR (100 MHz, $CDCl_3$) δ 140.62, 128.16, 127.76 (2 x C), 126.50 (2 xC), 122.14 (q, *J* = 275 Hz), 72.01, 70.77, 62.95, 32.72, 29.67, 29.38, 29.33, 26.08, 25.66.

## 8-[4-(3-Trifluoromethyl-3*H*-diazirin-3-yl)benzyloxy-octanoic acid (5, KK-242)]

Jones reagent (a few drops) was added to compound **4** (30 mg, 0.087 mmol) in acetone (5 mL) and the reaction was stirred for 2 hr. Excess Jones reagent was consumed by adding isopropyl alcohol (a few drops). The resulting blue solution was diluted with water (40 mL) and the product was extracted into ethyl acetate (30 mL x 3). The combined ethyl acetate extracts were washed with brine, dried over anhydrous $Na_2SO_4$, filtered and the solvent removed under reduced pressure on a rotary evaporator to give an oil which was purified by flash column chromatography (silica gel eluted with 20–50% ethyl acetate in hexanes) to give compound 5 (KK-242, 24 mg, 77%). [1]H NMR (400 MHz, $CDCl_3$) δ 7.37 (d, 2 H, *J* = 7.1 Hz), 7.19 (d, 2 H, *J* = 7.1 Hz), 4.51 (s, 2 H), 3.46 (t, 2 H, *J* = 6.2 Hz), 2.36 (t, 2 H, *J* = 6.1 Hz), 1.70–1.200 (m, 10 H); [13]C NMR (100 MHz, $CDCl_3$) δ 179.69, 140.56, 128.15, 127.78 (2 x C), 126.52 (2 x C), 122.14 (q, *J* = 275 Hz), 72.02, 70.68, 33.92, 29.60, 29.01, 28.93, 25.94, 24.57.

## Acknowledgements

We are grateful to Alex S Evers, Gustav Akk, and Joe Henry Steinbach for helpful discussions and comments on the manuscript. We are grateful to Stephen Traynelis for sharing the Channelab software.

## Additional information

### Funding

| Funder | Grant reference number | Author |
| --- | --- | --- |
| National Institutes of Health | R35GM137957 | Wayland WL Cheng |
| National Institutes of Health | F32GM139351 | John T Petroff |
| National Institutes of Health | R01HL067773 | Douglas F Covey |
| National Institutes of Health | R01GM108799 | Douglas F Covey |

The funders had no role in study design, data collection and interpretation, or the decision to submit the work for publication.

### Author contributions

Noah M Dietzen, Mark J Arcario, Data curation, Formal analysis, Writing – review and editing; Lawrence J Chen, Data curation, Formal analysis; John T Petroff, Data curation, Writing – review and editing; K Trent Moreland, Formal analysis; Kathiresan Krishnan, Methodology; Grace Brannigan, Formal analysis, Methodology, Writing – review and editing; Douglas F Covey, Conceptualization, Investigation, Methodology, Resources, Writing – review and editing; Wayland WL Cheng, Conceptualization, Data curation, Formal analysis, Funding acquisition, Investigation, Methodology, Resources, Supervision, Writing - original draft, Writing – review and editing

## Author ORCIDs

Mark J Arcario ![ORCID] http://orcid.org/0000-0001-5017-1519
John T Petroff II, ![ORCID] http://orcid.org/0000-0002-1323-0273
Wayland WL Cheng ![ORCID] http://orcid.org/0000-0002-9529-9820

## Decision letter and Author response

Decision letter https://doi.org/10.7554/eLife.74306.sa1
Author response https://doi.org/10.7554/eLife.74306.sa2

---

## Additional files

### Supplementary files
• Transparent reporting form

### Data availability
Figure 4—source data 1 contains the numerical data used to generate Figure 4A and 4B. Figure 4—source data 2 contains the numerical data used to generate Figure 4C and Figure 4—figure supplement 5. Figure 4—source data 3 contains the statistical analysis (linear mixed effects model) for Figure 4—figure supplement 5.

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
