## [Editor Report]

The authors use a combination of photolabeling and mass spectrometry to probe polyunsaturated fatty acid (PUFA) binding site locations in the pentameric ligand-gated ion channel (pLGIC) ELIC. The data strongly support the idea that DHA, but not PA, bind in the transmembrane domains of ELIC in two locations that overlap with that previously shown for the homolog GLIC. They also show that coarse-grained simulations can recapitulate the observation that DHA and not PA bind in this region, supporting the idea that such simulations can be useful for studying PUFA interactions with pLGICs. Strikingly, the authors provide evidence that DHA binding depends on the occupancy of the agonist site, which is an important observation that informs on molecular motions in the transmembrane domains in response to agonist binding. This work contributes to understanding the molecular underpinnings of PUFA modulation in pLGICs.

---

## [Decision Letter]

**Decision letter after peer review:**

Thank you for submitting your article "Polyunsaturated fatty acids inhibit a pentameric ligand-gated ion channel through one of two binding sites" for consideration by *eLife*. Your article has been reviewed by 3 peer reviewers, including Marcel P Goldschen-Ohm as the Reviewing Editor and Reviewer #1, and the evaluation has been overseen by Richard Aldrich as the Senior Editor. The following individuals involved in review of your submission have agreed to reveal their identity: Pierre-Jean Corringer (Reviewer #2); Michaela Jansen (Reviewer #3).

Essential revisions:

1) For all plots with mean +/- SEM including dose-response relations and bar plots, please show all of the individual data points. Also report all statistical tests used in the figure legends.

2) Regarding KK-242, similar chemistry and compounds have been described previously (for review see Chemical Reviews 2013 113 (10), 7880-7929). It should be clarified whether the compound described herein is a new chemical entity or whether it has been generated previously (Beilstein search?).

3) Given that ultimately the authors suggest that the DHA site in ELIC is similar to that in GLIC, a figure that compares the observed binding site in the GLIC structural model and the proposed binding site(s) in ELIC would be very helpful to orient readers.

4) The observation that DHA inhibition depends on agonist site occupancy is very interesting. However, please address the following concerns regarding this data: i) At the concentrations used the authors are looking for a reduction in a signal that is already only ~1%. If these concentrations were necessary to observe the effect, please comment on why that is. ii) It is not clear that there is any reason to normalize the data shown in Figure 4C to the 0 PUFA conditions (in any event, it is not clear how they were normalized as the data don't seem to be from paired experiments). Unless there is a compelling reason otherwise, the data in Figure 4C should be plotted as raw labeling efficiencies (no normalization) and all data points in the absence of PUFA should be shown. iii) Given the variation in labeling efficiency reported in Figure 4-S3, n=3 data points is on the low side. If feasible, another couple of data points would likely help greatly to solidify this dataset, although we appreciate that these may be difficult experiments. iv) How were the data in Figure 4C analyzed? ANOVA?

5) The suggestion of two possible docking sites based on observed photolabeling at Q264 in M3 and C313 in M4 seems entirely dependent on the choice of docked conformations. However, it is not clear how representative the docked poses shown in Figure 4A and B are to the collection of docked poses obtained. Do the vast majority of docked poses cluster tightly around these two poses, or not? A figure illustrating this would be very useful. Also, it is difficult to judge how close the two docked orientations are as they are presented in separate images with different viewpoints. A single image showing both proposed sites on either side of M4 would be very helpful for visualizing what is being proposed in a larger context (e.g. something along the lines of how DHA and PLC are shown in Basak et al. Figure 3A).

6) Were peak responses of R117C-M before and after DTT incubation obtained from the same liposomes? If not, there is no control for expression, and controls to see whether DTT itself increases peak responses in control or in R123C-M are needed.

7) The logic by which the authors conclude that DHA modulates via a single site needs to be spelled out much more clearly. Presumably, because R117C-M has a PUFA like effect, this must be the site? But then why does DHA still modulate R117C-M to a similar extend as control and all other mutants? Also, why would the R118A substitution abolish DHA modification but not the methylester? Both remove essentially a single charge in this interacting pair that seems to be in salt bridge distance?

*Reviewer #1:*

The authors use photolabeling in combination with mass spectrometry to identify a binding site for the polyunsaturated fatty acid (PUFA) DHA in the pentameric ligand-gated ion channel ELIC. The identified site is similar to a structure of DHA bound to the homolog GLIC. Most strikingly, the authors show that DHA binding in ELIC is dependent on the agonist-bound state of the channel, which is an important observation that informs on molecular motions in the transmembrane domains in response to agonist binding. However, the data for the state-dependence of DHA binding, although suggestive, could benefit from either a clearer presentation or some additional investigation.

The authors use a powerful combination of photolabeling and MS to probe polyunsaturated fatty acid (PUFA) binding site locations in the pentameric ligand-gated ion channel (pLGIC) ELIC. The data strongly support the idea that DHA, but not PA, bind in the transmembrane domains of ELIC in a similar location to that previously shown for the homolog GLIC. They also show that coarse-grained simulations can recapitulate the observation that DHA and not PA bind in this region, supporting the idea that such simulations can be useful for studying PUFA interactions with pLGICs. Most strikingly, the authors provide evidence that DHA binding depends on the occupancy of the agonist site, which is an important observation that informs on molecular motions in the transmembrane domains in response to agonist binding. For me, this was the most interesting aspect of the paper. However, I also have some concerns about the evidence that is provided for this which need addressing (see below).

1. For all plots with mean +/- SEM including dose-response relations and bar plots, please show all of the individual data points. Also report all statistical tests used in the figure legends.

2. Given that ultimately the authors suggest that the DHA site in ELIC is similar to that in GLIC, a figure that shows this is a glaring omission. Please show a figure that compares the observed binding site in the GLIC structural model and the proposed binding site(s) in ELIC for comparison. In GLIC, DHA appears to not really extend much into the transmembrane domain. Thus, the similarity to KK-242 whose tail does extend down between M4 and M3 is not completely clear. This should be discussed.

3. The most striking advance for me is the observation that DHA binding depends on occupancy of the agonist site. Although the data presented for this are suggestive, I have some concerns:

i) Why was 10uM KK-242 used for DHA competition experiments? At this low concentration the labeling efficiency is really low to begin with. I am concerned about the reliability of measuring a reduction of a signal that is already only 1-2%, and that furthermore displays a fairly large range of deviation (Figure 4-S3). Does a higher K-242 concentration outcompete DHA?

ii) Perhaps I am misunderstanding the experiment, but I don't understand why the data in Figure 4C are normalized. I think they should simply be plotted as the observed labeling efficiencies in each condition and all data points in the absence of PUFA need to be shown. After the MS it's not like you can take that same sample and test it on another PUFA condition.

iii) Given the variation in labeling efficiency reported in Figure 4-S3, I am not completely convinced that 10uM DHA has much effect on labeling. Even though the effect at 30uM DHA looks fairly clear, I am still a bit concerned about having only three data points given such variability and low initial signals. I suggest obtaining at least another couple of data points as this is a crucial experiment regarding the state-dependent action of DHA.

iv) What is the statistical test indicated in Figure 4C? Probably should be ANOVA with some posthoc test.

4. The suggestion of two possible docking sites based on observed photolabeling at Q264 in M3 and C313 in M4 seems entirely dependent on the choice of docked conformations. However, it is not clear how representative the docked poses shown in Figure 4A and B are to the collection of docked poses obtained. Do the vast majority of docked poses cluster tightly around these two poses, or not? A figure illustrating this would be useful. In GLIC, DHA falls between two arginines that are similarly oriented to R117 and R123, so how unlikely is such an orientation in ELIC? Also, it is difficult to judge how close the two docked orientations are as they are presented in separate images with different viewpoints. A single image showing both proposed sites on either side of M4 would be very helpful for visualizing what is being proposed in a larger context (e.g. something along the lines of how DHA and PLC are shown in Basak et al. Figure 3A). Along these same lines, it is unclear to me that the photolabeling is not consistent with some flexibility in the tail position at a single site given that the photolabeled positions are quite close. If the photolabeling is truly strong evidence for two distinct sites, this needs to be explained further. Otherwise, I suggest presenting the data as consistent with two possible sites on either side of M4 analogous to DHA and PLC as observed in GLIC rather than direct evidence for two sites.

5. The logic of the final Results section on hMTS was a bit hard for me to follow. Firstly, hMTS mimics the PA tail (line 449), but PA does not bind here specifically? Second, no differences in peak current were observed between control and unmodified or modified cysteine mutants as assayed by ANOVA. Although I do agree that modification of R117C-M does seem like it exhibits smaller peak currents, I am a bit shaky on the idea of falling back to t-Test when ANOVA does not indicate a difference. More importantly, were peak responses of R117C-M before and after DTT incubation obtained from the same liposomes? If not, there is no control for expression, and DTT itself was not checked to see if it increases peak responses in control or in R123C-M. Thus, the evidence that hMTS inhibits ELIC currents analogous to a PUFA, although suggestive, is lacking controls. Regardless, the logic that DHA modulates via a single site is not spelled out sufficiently for me. Presumably, because R117C-M has a PUFA like effect, this must be the site? But then why does DHA still modulate R117C-M to a similar extend as control and all other mutants?

*Reviewer #2:*

This is a very compelling study identifying a fatty acid binding site mediating allosteric inhibition of ELIC. The study implements several complementary techniques that are all relevant to the demonstration:

1. Purification of ELIC and reconstitution in giant liposomes for excised patch voltage-clamp, showing that DHA inhibits ELIC channel function.

2. Design and synthesis of a new photolabeling reagent, KK-242, with optimal photochemistry with a structure resembling a fatty acid. Middle-down mass spectrometry show that KK-242 photolabels two distinct sites on both sides of the outer portion of M4, one at the interface with M3 (M3Q264 and M4R318), and one at the interface with M1 (M4C313). Importantly, KK-242 labelling is inhibited by DHA in a dose-dependent manner and only in the presence of the agonist. Thus, DHA preferentially binds to both KK-242 photolabeled sites when ELIC is in the agonist-bound state.

3. Using coarse-grained molecular dynamics (CGMD) simulations, authors determine two fatty acid binding sites in the outer TMD of ELIC that are specific for DHA over PA when agonist is bound to the channel, and that are overlapping with the one identified by photolabelling.

4. To investigate the functional significance of the M1 and M3 DHA binding sites, each site was covalently modified with hexadecyl-methanethiosulfonate (hMTS), mimicking the binding of a fatty acid, using the accessible arginines above the M1M4 and M3M4 groves. Electrophysiological analysis of ELIC cysteine mutants clearly show that the M3M4 site specifically mediates inhibition.

Overall, the study is carefully designed and conducted, very clearly presented and discussed, and the demonstration complete. It is a beautiful piece of work.

*Reviewer #3:*

Dietzen et al. investigated the interaction site/s and mechanism of the polyunsaturated fatty acid (PUFA) docosahexaenoic acid (DHA) with the prokaryotic pentameric ion channel ELIC using a novel fatty acid photolabeling reagent with broad amino acid reactivity coupled with mass spectrometry, coarse-grained molecular dynamics (CGMD) calculations, as well as covalent modification using a methane thiosulfonate (MTS) reagent with a lipid-like tail of engineered Cys at identified sites. The results are further probed using a DHA methylester derivative (DHA-ME), using competition photolabeling with palmitic acid (PA), and current simulations based on a previously published gating model.

The authors show that DHA or DHA-ME preapplication for 3 minutes reduces the peak current obtained with the agonist cysteamine. The cysteamine EC50 is unaltered. PA has a significant but much smaller effect. CGMD was used to calculate lipid diffusion, and identified localized areas of high enrichment for DHA in two intrasubunit groves between M4 and M1 or M3, respectively, mostly in the outer membrane leaflet. The distribution of PA was also calculated with the same method and found to be more diffuse, indicating that PA's functional effect is associated with unspecific rather than specific interactions.

In the next set of experiments the goal was to use photoaffinity labeling with suitable fatty acids and subsequent mass spectrometric identification to characterize PUFA binding sites. Initially, a commercially available bifunctional photoreactive reagent (pacFA) with an aliphatic diazirine was used. The authors find that PacFA did not label ELIC, likely because the hydrophobic transmembrane environment does not contain nucleophilic amino acid sidechains like glutamate or aspartate that are reactive towards the aliphatic diazirine. The authors subsequently design and synthesize the novel trifluoromethylphenyl diazirine compound, KK-242, that is expected to be more broadly reactive towards different amino acid side chains. KK-242 has a similar alkyl chain length compared to PA. In photolabeling experiments with KK-242 this compound labeled a single position in M3 (Q264), and two positions in M4 (C313 and R318). Docked in chemically meaningful poses KK-242 seems to bind to two distinct sites on either side of the outer portion of M4. The KK-242 carboxylate reaches to R117 when docked to Q264, and to R123 when docked to C313.

Previously, a co-crystal structure of GLIC with DHA was published (Basak et al., *eLife* 2017). Both GLIC and ELIC contain an Arg at the position corresponding to ELIC-R117. GLIC contains an additional Arg, R118, whereas ELIC contains an Arg at position 123, R123. Interestingly, the GLIC structure had the DHA carboxylate in close proximity to R118. The prior GLIC study showed that a R118A mutation is not inhibited by DHA, indicating that the R118, and consequently potentially a salt bridge between the DHA carboxylate and the R118 positive charge was required for DHA inhibition. On the contrary, the present study indicates that the methyl ester of DHA, DHA-ME, is able to inhibit ELIC. In this case, there would be no salt bridge between DHA-ME and Arg. The photolabeling was further interrogated with competition experiments with DHA and PA. These showed that photolabeling efficiency by KK-242 was dose dependently reduced for the M4 site with DHA but only insignificantly reduced with PA. CGMD simulation data confirms specific binding at M3/M4 intrasubunit grove for DHA, same site as R117, Q264. Additional binding with the M1/C313 is also identified. In summary, the photolabeling results are supported by the CGMD simulations.

In a separate set of experiments, in a Cys-less background R117C and R123C were modified with a MTS reagent with a lipid-like tail for investigation of the functional significance of sites. R117C modification with MTS reagent inhibited peak currents and the effect was reversed with reducing agent application (dithiothreitol, DTT) which will remove the covalently linked disulfide introduced by the MTS reagent. There was no effect of the covalent modification of R123C with the same reagent. Mass spectrometry for both positions indicated that both R117C and R123C were indeed modified at a comparable high level. This leads to the conclusion that while both positions are modified, only the R117C modification has a functional impact that can be observed with the electrophysiological experiments described here.

The authors further show that photolabeling was specific for the agonist bound state.

Simulations previously published by others and the present study also indicate that fatty acids in the presence of agonist may stabilize a pre-active or desensitized state of the channel.

In summary, the present study uses complementary approaches that identify two binding sites for the PUFA DHA within the extracellular leaflet on either side of the M4 transmembrane segment, with only a single site per subunit being responsible for the inhibitory effect.

Overall, the results recapitulate what has been described and published in *eLife* in 2017 with regard to DHA modulation of the closely related pentameric channel homologue GLIC using X-ray co crystallography, EPR and electrophysiology. The present paper uses a different set of approaches that are elegantly complementary to one another and overall corroborate the same findings. Of note, the GLIC co-crystal structure did show two lipid binding sites as well, one was occupied by DHA, the other by "PLC", a lipid carried with the purification. The newly-synthesized compound KK-242 is a promising tool likely attractive for the study of additional transmembrane proteins that are modulated by PUFAs.

Comments for the authors:

Similar chemistry and compounds have been described previously (for review see Chemical Reviews 2013 113 (10), 7880-7929), it should be clarified whether the compound described herein is a new chemical entity or whether it has been generated previously (Beilstein search?).

Statistics: the detailed results should be provided.

Why would the R118A substitution abolish DHA modification but not the methylester? Both remove essentially a single charge in this interacting pair that seems to be in salt bridge distance?

---

## [Author Response]

Reviewer #1:The authors use photolabeling in combination with mass spectrometry to identify a binding site for the polyunsaturated fatty acid (PUFA) DHA in the pentameric ligand-gated ion channel ELIC. The identified site is similar to a structure of DHA bound to the homolog GLIC. Most strikingly, the authors show that DHA binding in ELIC is dependent on the agonist-bound state of the channel, which is an important observation that informs on molecular motions in the transmembrane domains in response to agonist binding. However, the data for the state-dependence of DHA binding, although suggestive, could benefit from either a clearer presentation or some additional investigation.The authors use a powerful combination of photolabeling and MS to probe polyunsaturated fatty acid (PUFA) binding site locations in the pentameric ligand-gated ion channel (pLGIC) ELIC. The data strongly support the idea that DHA, but not PA, bind in the transmembrane domains of ELIC in a similar location to that previously shown for the homolog GLIC. They also show that coarse-grained simulations can recapitulate the observation that DHA and not PA bind in this region, supporting the idea that such simulations can be useful for studying PUFA interactions with pLGICs. Most strikingly, the authors provide evidence that DHA binding depends on the occupancy of the agonist site, which is an important observation that informs on molecular motions in the transmembrane domains in response to agonist binding. For me, this was the most interesting aspect of the paper. However, I also have some concerns about the evidence that is provided for this which need addressing (see below).1. For all plots with mean +/- SEM including dose-response relations and bar plots, please show all of the individual data points. Also report all statistical tests used in the figure legends.

Done.

2. Given that ultimately the authors suggest that the DHA site in ELIC is similar to that in GLIC, a figure that shows this is a glaring omission. Please show a figure that compares the observed binding site in the GLIC structural model and the proposed binding site(s) in ELIC for comparison. In GLIC, DHA appears to not really extend much into the transmembrane domain. Thus, the similarity to KK-242 whose tail does extend down between M4 and M3 is not completely clear. This should be discussed.

We agree that we did not adequately discuss the similarities and differences between our assessment of DHA binding in the M3/M4 site and the GLIC structure. We have added Figure 6—figure supplement 3 to provide a comparison of the M3/M4 DHA binding site in ELIC and the GLIC structure complexed with DHA. Our assessment of DHA binding in ELIC is coarse-grained and the simulations do not provide just a single binding mode. Therefore, to achieve the most accurate comparison between our results and the GLIC structure, we have overlaid the atomic structure of ELIC, color-coded with the highest DHA contact probabilities, with the structure of GLIC complexed with DHA. We also provide a sequence alignment of high contact probability residues in ELIC and the analogous residues in GLIC. As noted by the reviewer, the interaction of DHA in the M3/M4 site in ELIC is similar to the site identified in the crystal structure but not the same. The polyunsaturated tail of DHA in the GLIC structure is not fully resolved and appears to be extending into the membrane. In contrast, our simulations show that the highest contact probabilities are of the polyunsaturated tail interacting with hydrophobic residues such as A268 and G271 in the M3/M4 groove. The reason for this discrepancy with the GLIC crystal structure is unclear. It may be a consequence of the two contrasting methodologies or a real difference in DHA interactions between these two pLGICs. Nevertheless, we argue that the DHA binding site we describe is plausible and functionally relevant because the polyunsaturated tail is critical for DHA binding affinity and efficacy (see discussion below regarding this specific issue). We have added content in the Discussion to emphasize the similarities and differences between the interactions of DHA with ELIC and the GLIC-DHA crystal structure.

3. The most striking advance for me is the observation that DHA binding depends on occupancy of the agonist site. Although the data presented for this are suggestive, I have some concerns:i) Why was 10uM KK-242 used for DHA competition experiments? At this low concentration the labeling efficiency is really low to begin with. I am concerned about the reliability of measuring a reduction of a signal that is already only 1-2%, and that furthermore displays a fairly large range of deviation (Figure 4-S3). Does a higher K-242 concentration outcompete DHA?

By raising the issue of KK-242 concentration, the reviewer has identified the primary challenge associated with performing competition photolabeling experiments. It is necessary to use a low concentration of photolabeling reagent, typically lower than the concentration of the competitor (unless the competitor has a much higher affinity). This is because photolabeling is an irreversible reaction, and the photolabeling reagent must have a relatively low occupancy of the binding site for a competitor to significantly reduce the labeling efficiency. We have empirically found from our studies with neurosteroids and other lipids that 10uM or less of photolabeling reagent is necessary to observe competition with equivalent or higher concentrations of competitor. This is the rationale for the concentration of KK-242 used.

Regarding the issue of low labeling efficiency, the mass spectrometer has a large dynamic range (~1E6). Therefore, what matters to achieve an accurate measurement, with sufficient signal to noise, is not the labeling efficiency per se but the absolute signal of labeled and unlabeled peptides. To show the signal that we are obtaining, we have included the source data for Figure 4C (Figure 4- source data 2), which shows the exact MS peptide (MS1) intensities for each replicate from the competition experiments. For the Orbitrap mass spectrometer, signals in the e3 range are approaching the noise threshold of the instrument and may yield inaccurate measurements. Note that the lowest intensities obtained for labeled peptides (replicate 1) in the competition experiment are still in the mid-e4 range well above instrument noise. Also note that the intensities for unlabeled peptides are in the e6-e8 range, thus yielding efficiencies even below 1%. In other studies, we have accurately measured labeling efficiency as low as 0.1% when the signal for the unlabeled peptide is very high.

Indeed, the labeling efficiencies in Figure 4-S3 display high variability. We have now plotted the un-normalized labeling efficiencies for each replicate of the competition experiments in Figure 4-S5. Note that these labeling efficiencies, which are from three separate replicates, show even greater variability. We will discuss the interpretation of these data and the reason for the variability below, in response to whether the competition data should be normalized.

ii) Perhaps I am misunderstanding the experiment, but I don't understand why the data in Figure 4C are normalized. I think they should simply be plotted as the observed labeling efficiencies in each condition and all data points in the absence of PUFA need to be shown. After the MS it's not like you can take that same sample and test it on another PUFA condition.

The data in Figure 4C should be normalized because these are indeed groups/replicates of samples. Each replicate of samples is analyzed in a single queue by LC-MS/MS. Importantly, a single queue (continuous run of samples) uses the same reverse phase column and electrospray ionization emitter tip. The column and emitter tip are frequently replaced (the emitter tip is replaced after almost each queue), and we have found that differences between columns or emitter tips (which are both made in-house) lead to significant differences in peptide intensities and labeling efficiencies. This may be because the unlabeled and labeled peptides elute at different times in the acetonitrile gradient and the ionization efficiencies vary dramatically at different acetonitrile concentrations with different emitter tips. However, within a single queue of samples (i.e. a single replicate) using the same column and emitter, the data are very consistent. To illustrate this, we plot the unnormalized labeling efficiencies of each of the three replicates from Figure 4C in Figure 4-S5. We have also included the MS peptide intensities in Figure 4- source data 2 for full transparency. While the absolute labeling efficiencies from one replicate to another vary significantly (Figure 4-S5), the relative labeling efficiencies within a single replicate are comparable (Figure 4C). Therefore, we believe the data is best visualized when normalized within a single replicate.

While we define “labeling efficiency” as the ratio of labeled peptide to total peptide, this value is a specific construct the middle-down MS approach rather than a true labeling efficiency, since we are not obtaining absolute measurements of peptide abundance from the MS data (i.e. MS intensity is a function of peptide abundance and ionization efficiency). Therefore, we do not ascribe significant meaning to absolute labeling efficiency values. Of note, the intact protein MS measurements should provide a more accurate measurement of labeling efficiency at the subunit level, since unlabeled and labeled ELIC intact subunits are expected to have similar ionization efficiencies and the entire sample is sprayed simultaneously through direct injection. To clarify this, we have changed “labeling efficiency” to “apparent labeling efficiency” for the middle-down MS data in the Results section and added content regarding the origin of the variability in apparent labeling efficiencies from one replicate to another.

iii) Given the variation in labeling efficiency reported in Figure 4-S3, I am not completely convinced that 10uM DHA has much effect on labeling. Even though the effect at 30uM DHA looks fairly clear, I am still a bit concerned about having only three data points given such variability and low initial signals. I suggest obtaining at least another couple of data points as this is a crucial experiment regarding the state-dependent action of DHA.

Considering the above discussion about replicates, we have plotted normalized data in Figure 4C to clearly illustrate the effect of DHA on labeling efficiency in the presence of agonist. We cannot perform a one-way ANOVA when the data is normalized. To test for differences between conditions, we analyzed the unnormalized data using a linear mixed effects model (Figure 4- source data 3). By accounting for fixed effects (i.e. adding DHA or PA, and cysteamine) and the random effect of variation between replicates, we find that, when cysteamine is present, the effect of 10 and 30 μm DHA in competitively preventing KK-242 labeling is statistically significant (p = 0.016). The results of this analysis are now shown in Figure 4-S5 and Figure 4- source data 3. With three replicates, this result is robust, and we argue that additional replicates are not necessary. Previous studies using such competition experiments routinely conduct three replicates (Li et al. 2010. JBC. 285(12):8615-8620; Jayakar. 2019. Mol Pharm. 95(6): 615-628). By using two different concentrations of DHA or PA and performing three replicates, the design of our experiment meets the standard within the field for this approach.

iv) What is the statistical test indicated in Figure 4C? Probably should be ANOVA with some posthoc test.

We have indicated in the legend for Figure 4 that the data were analyzed using a linear mixed effects model. The results for this analysis are shown in Figure 4-S5 and Figure 4- source data 3.

4. The suggestion of two possible docking sites based on observed photolabeling at Q264 in M3 and C313 in M4 seems entirely dependent on the choice of docked conformations. However, it is not clear how representative the docked poses shown in Figure 4A and B are to the collection of docked poses obtained. Do the vast majority of docked poses cluster tightly around these two poses, or not? A figure illustrating this would be useful. In GLIC, DHA falls between two arginines that are similarly oriented to R117 and R123, so how unlikely is such an orientation in ELIC? Also, it is difficult to judge how close the two docked orientations are as they are presented in separate images with different viewpoints. A single image showing both proposed sites on either side of M4 would be very helpful for visualizing what is being proposed in a larger context (e.g. something along the lines of how DHA and PLC are shown in Basak et al. Figure 3A). Along these same lines, it is unclear to me that the photolabeling is not consistent with some flexibility in the tail position at a single site given that the photolabeled positions are quite close. If the photolabeling is truly strong evidence for two distinct sites, this needs to be explained further. Otherwise, I suggest presenting the data as consistent with two possible sites on either side of M4 analogous to DHA and PLC as observed in GLIC rather than direct evidence for two sites.

We suggest that two “sites” are photolabeled based on two findings: (1) labeling of ELIC by KK-242 shows both singly and doubly-labeled subunits as assessed by intact protein mass spectrometer (Figure 3B), and (2) multiple residues in M4 facing either M1 or M3 are labeled. The intact protein MS experiment, which shows ELIC subunits photolabeled with two KK-242, indicates that there are at least two non-overlapping sites that are labeled per subunit. However, we agree that we cannot rule out the possibility that this labeling is occurring from two KK-242 molecules binding to just one side of M4. We have modified the language in the Results section to convey that the data is suggestive of two sites similar to what was observed in GLIC.

Regarding the docking results, the primary purpose of this work was to demonstrate that it is geometrically plausible that KK-242 label the identified residues via the photoreactive diazirine group, when oriented as expected in the amphipathic environment of the lipid bilayer. Therefore, we intentionally showed just one pose in Figure 4A and 4B to make this point. We rely instead on the coarse-grained MD simulations to examine fatty acid binding sites more rigorously in the context of a lipid bilayer. Molecular docking has limited utility in identifying binding poses for long flexible lipid-like molecules, especially since the lipid bilayer is not considered. Not surprisingly, the docking results yielded multiple distinct poses, some of which are less plausible in a lipid bilayer. To illustrate the diversity of poses obtained, we clustered the docking poses using an RMSD of 5 Å and displayed representative poses from the top 5 most abundant clusters (Figure 4- S4A and S4B). As requested, we have also included a perspective of both putative KK-242 binding poses illustrating the two sites on either side of M4 (Figure 4- S4C).

5. The logic of the final Results section on hMTS was a bit hard for me to follow. Firstly, hMTS mimics the PA tail (line 449), but PA does not bind here specifically? Second, no differences in peak current were observed between control and unmodified or modified cysteine mutants as assayed by ANOVA. Although I do agree that modification of R117C-M does seem like it exhibits smaller peak currents, I am a bit shaky on the idea of falling back to t-Test when ANOVA does not indicate a difference. More importantly, were peak responses of R117C-M before and after DTT incubation obtained from the same liposomes? If not, there is no control for expression, and DTT itself was not checked to see if it increases peak responses in control or in R123C-M. Thus, the evidence that hMTS inhibits ELIC currents analogous to a PUFA, although suggestive, is lacking controls. Regardless, the logic that DHA modulates via a single site is not spelled out sufficiently for me. Presumably, because R117C-M has a PUFA like effect, this must be the site? But then why does DHA still modulate R117C-M to a similar extend as control and all other mutants?Before explaining the logic of these experiments, here are a few points of clarification about the results.

1) Control (C300S/C313S), R117C (C300S/C313S/R117C), R117C-M (C300S/C313S/R117C modified with hMTS), and R123C-M (C300S/C313S/R123C modified with hMTS) data are collected from different liposome preps containing the indicated ELIC proteins. However, the R117C-M and R117C-M DTT data are obtained from the same liposomes- that is, liposomes made with R117C-M. These recordings were performed either in the absence or presence of DTT. Therefore, while we performed an ANOVA analysis to compare all conditions, it is also valid to perform a T-test between these two samples only. Indeed, we believe that this comparison is the most critical one, which demonstrates that the presence of the hexadecyl group significantly inhibits ELIC responses. We have deleted content for T-test analyses of sample pairs other than R117C-M and R117C-M DTT, which we acknowledge are inappropriate.

2) DHA inhibits R117C-M currents to the same extent as R117C, R123C-M and control. However, as we note in the Discussion, R117C-M currents were very small; on average, <10% of R117C currents. We suggest in the Discussion section that these residual currents may arise from a small amount of unmodified R117C within the R117C-M prep. This small amount of unmodified R117C may have been undetected in the intact protein MS experiment because of the high amount of lipid and other adducts that increase the noise in the intact protein mass spectrum (Figure 6- S1A). The other intact protein MS analyses of hexadecyl-MTS modified proteins were of DDM samples that contained much less lipid and had minimal noise. Other possible explanations for this result are stated in the Discussion. Importantly, the fact that R117C is inhibited by DHA to the same extent as control indicates that R117 is not critical for the affinity or efficacy of DHA (we will compare this result with the GLIC study below, in response to Reviewer #3).

Considering the above points, the logic of the hMTS experiment is as follows. Since the alkyl tail of DHA is likely the most important determinant of its inhibitory effect, we reasoned that introducing a long aliphatic group at the identified sites may mimic this inhibitory effect. We recognize that a saturated hexadecyl group mimics PA but is not the same as the polyunsaturated tail of DHA. However, we are limited to what is commercially available for MTS reagents. Since PA weakly inhibits ELIC, it is still reasonable to expect that a hexadecyl group at this site will inhibit the channel to some extent. Indeed, modification of R117C but not R123C with hexadecyl-MTS inhibits ELIC responses significantly. Why does the hexadecyl group at residue position 117 inhibit ELIC responses to a greater extent than PA alone? We suggest in the Discussion section that the weak effect of PA is due to its low binding affinity. However, if a PA-like molecule (hexadecyl MTS) is introduced at the inhibitory site with 100% occupancy (i.e. chemical modification), it strongly inhibits channel responses. Overall, the results suggest that occupancy of the M3/M4 groove mediates the inhibitory effect of fatty acids.

Reviewer #3:Comments for the authors:Similar chemistry and compounds have been described previously (for review see Chemical Reviews 2013 113 (10), 7880-7929), it should be clarified whether the compound described herein is a new chemical entity or whether it has been generated previously (Beilstein search?).

We have searched for this compound in Chemical Abstracts and confirmed that it has not been previously reported. KK-242 is indeed a newly described chemical entity. However, we are aware that there are similar compounds previously reported. For example, Quay et al. and Capone et al. described fatty acid analogs with a diazirinephenoxy or azidophenoxy group (Quay et al. 1982. JBC. 256(9): 444-4449; Capone et al. 1983. JBC. 258(3): 1395-1398). In the Introduction, we stated that “fatty acid photolabeling reagents have been used previously to identify fatty acid binding proteins, but not to map binding sites at the residue level”. In that sentence, we referenced these studies.

Statistics: the detailed results should be provided.

The type of statistical analysis for the data in each figure have now been included in all figure legends.

Why would the R118A substitution abolish DHA modification but not the methylester? Both remove essentially a single charge in this interacting pair that seems to be in salt bridge distance?

We tested an R117C mutant (R117C/C300S/C313S) in ELIC and found that it had no effect on DHA inhibition compared to C300S/C313S (control) or WT (i.e. DHA still inhibits this mutant). In contrast, modification of R117C with hexadecyl-MTS (R117C-M) inhibits channel responses. As discussed above in response to Reviewer #1, we observed very small residual currents in R117C-M that are still sensitive to DHA leading us to tentatively conclude that these come from a small amount of unmodified R117C. See the above response for more details regarding this.

Irrespective of the R117C-M results, the fact that unmodified R117C is still sensitive to DHA and that DHA methyl ester also inhibits ELIC strongly suggest that a salt bridge interaction between DHA and a basic amino acid is not critical for DHA binding or effect. This is consistent with the CGMD simulations, which show that DHA has a much higher affinity than PA even though both molecules are modeled to have an anionic carboxylate headgroup. This indicates that the polyunsaturated tail is the primary determinant of binding affinity. We have added content in the Discussion to elaborate on this .

The reviewer may also be referring to the R118A mutation, which was made in GLIC, and shown to reduce the effect of DHA on desensitization (Basak et al. 2017. *eLife*. 6:e23886). Some differences between the GLIC experiment and the present study are: (1) recordings were made by two-electrode voltage clamp in whole oocytes with co-application of DHA with agonist, (2) the effect of DHA was assessed using steady-state current relative to peak, and (3) the arginine was mutated to an alanine instead of a cysteine. With the GLIC experiments, it is possible that a relatively slow solution change in oocytes led to sub-optimal measurements of the effect of DHA on steady-state:peak currents. In addition, co-application of agonist and DHA may differentially affect peak currents in WT versus mutant. In contrast, we monitored the effect of pre-application of DHA on peak currents in excised patches with rapid application of agonist (the primary effect of DHA previously reported in other pLGICs) and find that R117 in ELIC (equivalent to R118 in GLIC) is not important for the inhibitory effect of DHA. In addition to differences in methodology between the GLIC study and the present work, it is also possible that the mechanism of DHA inhibition in GLIC differs from ELIC. However, the characteristics of PUFA inhibition in many pLGICs appears to be generally conserved. We argue against the importance of a salt bridge interaction in mediating PUFA binding and inhibition of pLGICs , although this remains to be tested in pLGICs other than ELIC.